



# An observational record of global gridded near surface air temperature change over land and ocean from 1781

Colin P. Morice [1], David I. Berry[2], Richard C. Cornes[2], Kathryn Cowtan[3], Thomas Cropper[2], Ed Hawkins[4], John Kennedy[1], Timothy J. Osborn[5], Nick A. Rayner[1], Beatriz Recinos Rivas[6,2], Andrew P. Schurer[6], Michael Taylor[5], Praveen R. Teleti[4], Emily J. Wallis[5], Jonathan Winn[1], and Elizabeth C. Kent[2]

[1]Met Office, Exeter, EX1 3PB, United Kingdom
[2]National Oceanography Centre, Southampton, SO14 3ZH, United Kingdom
[3]University of York, York, YO10 5DD, United Kingdom
[4]National Centre for Atmospheric Science, Department of Meteorology, University of Reading, Reading. UK.
[5]Climatic Research Unit, School of Environmental Sciences, University of East Anglia, Norwich, NR4 7TJ, United Kingdom
[6]University of Edinburgh, School of Geosciences, Edinburgh, EH9 3JW, United Kingdom

**Correspondence:** Colin P. Morice (colin.morice@metoffice.gov.uk)

**Abstract.** We present a new gridded data set of air temperature change across global land and ocean extending back to the 1780s. This data set, called the GloSAT reference analysis, has two novel features: it uses marine air temperature observations rather than the sea surface temperature measurements typically used by pre-existing data sets, and it extends further into the past than existing merged land and ocean instrumental temperature records which typically estimate temperature changes from the mid-to-late 19th century onwards. New estimates of diurnal heating biases in marine air temperatures have enabled the use of daytime observations, extending the dataset further into the past compared to nighttime-only marine air temperature data. The data set uses an extended version of the CRUTEM5 station database over land areas, incorporating newly available bias adjustments for non-standard thermometer enclosures used prior to the adoption of Stevenson screens and new climatological normal estimates for stations with limited data in the 1961-1990 baseline period. Land and marine temperature anomalies are combined to produce a gridded data set following the methods developed for HadCRUT5. The GloSAT global and hemispheric temperature anomaly series show close agreement with those based on sea-surface temperature for much of the overlapping period of their records but with slightly less warming overall.



# 1 Introduction

Instrumental data sets recording changes and variations in near-surface temperature across the globe have been widely used to monitor changes in global and regional climate (World Meteorological Organisation, 2023). The Intergovernmental Panel on Climate Change 6th Assessment Report (IPCC AR6) assessment of global average temperature change from the mid-19th century (Gulev et al., 2021) is underpinned by instrumental data sets that combine information from Sea Surface Temperature (SST) observations with information from meteorological station observations of Land Surface Air Temperature (LSAT). The IPCC AR6 refers to change in mean near-surface temperature based on this combination of SST and LSAT as Global Mean Surface Temperature (GMST). Studies using climate model output commonly assess changes in near-surface temperature using air temperature changes over both land and ocean, using marine air temperature (MAT) rather than SST. The IPCC AR6 refers to this combination of LSAT and MAT as Global Surface Air Temperature (GSAT). Hence, the difference between GMST and GSAT is the use of SST or MAT to measure changes over the marine regions.

This article describes a new gridded GSAT data set. This data set has been constructed using a workflow in common with the existing HadCRUT5 GMST data set (Morice et al., 2021). We call this new data set the GloSAT reference analysis (GloSATref.1.0.0.0). It combines an extended and improved version of the CRUTEM5 LSAT data base, updated from Osborn et al. (2021), with a new marine MAT data set based on all-hours observations, building on Cornes et al. (2020). Section 2 summarises gridded datasets used in currently available observational GMST data sets. Section 3 describes the processing of input LSAT and MAT data sets and their use to construct the GloSAT reference analysis. Section 4 presents global and regional climate diagnostics derived from the GloSAT reference analysis. Conclusions are presented in section 5.

# 2 Background summary of existing instrumental records for GMST

Existing global data sets used to monitor changes in GMST combine measurements of near-surface air temperatures at meteorological stations (LSATs) with measurements of surface water temperatures obtained by ships, buoys, and moorings (SSTs). The currently available data sets of this type include HadCRUT5 (Morice et al., 2021), GISTEMP (Lenssen et al., 2019), NOAAGlobalTemp (Huang et al., 2022), Berkeley Earth (Rohde and Hausfather, 2020), Kadow et al. (Kadow et al., 2020), CMST 2.0 (Sun et al., 2022), DCENT (Chan et al., 2024) and Calvert (Calvert, 2024). Each of these data sets provides global grids that map GMST variability and change along with derived global and regional average time series. These data products have starting dates ranging from 1850 to 1880 as data availability, particularly of SST, decreases rapidly prior to this.

Data sources for current observational surface temperature data sets are shown in Table 1. There is much common observational data underpinning the named LSAT and SST data sets because of global data sharing and consolidation of national records into global archives. For example, the International Comprehensive Ocean-Atmosphere Data Set (ICOADS) (Freeman et al., 2018) is an underpinning source for each of the marine data sets in Table 1. Despite these overlaps in observation sources there are differences in selection criteria used in each data set and sometimes standard data holdings are extended through, e.g., addition of newly digitised historical sources. Each data set also differs in some or all of methods for data quality control (QC), methods used to account for systematic changes in observing practices, gridding methods and uncertainty modelling.

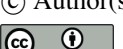



**Table 1.** Global data sets reporting GMST and their land and marine data sources. LSAT data sets: CRUTEM5 (Osborn et al., 2021); GHCNm v4 (Menne et al., 2018); Berkeley Earth land record (Rohde et al., 2013a, b); C-LSAT 2.0 (Li et al., 2021); DCLSAT v1.0 (Chan et al., 2024). SST data sets: HadSST4 (Kennedy et al., 2019); ERSST v5 (Huang et al., 2017); DCSST v1.0 (Chan et al., 2024). All land data sources are LSAT data sets. All marine data sources are SST data sets except for GloSATMAT, used in GloSATref, which is an all-hours MAT data set.

| Global data set | Land data source | Marine data source |
|---|---|---|
| HadCRUT5 | CRUTEM5 | HadSST4 |
| GISTEMP v4 | GHCNm v4 | ERSST v5 |
| NOAAGlobalTemp v6 | GHCNm v4 | ERSST v5 |
| Berkeley Earth | Berkeley Earth land record | HadSST4 |
| Kadow et al. (2020) | CRUTEM5 (via HadCRUT5) | HadSST4 (via HadCRUT5) |
| CMST 2.0 | C-LSAT 2.0 | ERSST v5 |
| DCENT v1.0 | DCLSAT v1.0 | DCSST v1.0 |
| Calvert (2024) | CRUTEM5 (via HadCRUT5) | HadSST4 (via HadCRUT5) |
| GloSATref | GloSATLAT | GloSATMAT (MAT) |

Some data sets have independent processing workflows, such as NOAAGlobalTemp v6 and HadCRUT5. Others share aspects of processing, such as the reprocessing of HadCRUT5 data using alternative gridding methods by Kadow et al. (2020) and alternative gridding methods and bias adjustment in Calvert (2024). The set of eight currently updated GMST data sets each use a combination of one of five LSAT data sets and one of three SST data sets (Table 1).

Historically, SST rather than MAT has been used in global temperature products for three main reasons (Jones et al., 1999): 1)
there are substantially fewer MAT observations available since ca. 1900; 2) diurnal heating biases in the MAT data that require adjustment; and 3) adjustment of MAT measurements to a common reference height to account for changes in measurement heights throughout the MAT record. SST observations are also affected by sampling limitations and systematic changes in measurement practices (Kennedy et al., 2019; Kent and Kennedy, 2021), but the adjustments required for MAT records were considered more complex and uncertain (Rayner et al., 2003). Furthermore, while a decline in ship coverage has resulted in
fewer observations of both SST and MAT since the 1960s, in the case of SST this has been mitigated through the increasing coverage of drifting buoy and satellite data (Kent and Kennedy, 2021).

Nighttime Marine Air Temperature (NMAT) observations have previously been used to avoid the need to make diurnal heating bias corrections and are available from CLASSnmat (Cornes et al., 2020) and UAHNMATv1 (Junod and Christy, 2020). Both ERSST v5 and HadSST4 use NMAT observations within their SST bias adjustment methods. ERSST v5 uses NMAT-SST
differences to derive a bias adjustment applied to ship SST observations prior to 2010. The HadSST4 bias adjustment model





uses NMAT-SST differences to constrain estimates of contributions of different measurement types for observations made from 1850 to 1920. These NMAT data sets have not previously been combined with LSAT to create GSAT data sets.

The IPCC AR6 (Gulev et al., 2021) reviewed whether differences between GMST and GSAT reconstructions should be expected. This included a review of a range of model-based studies and process understanding and differences between instrumental NMAT and SST data sets. Differences between the two diagnostics have been the subject of recent debate due to the common (but not exclusive) use of GSAT to assess near surface temperature changes in climate models while observational records have been based on GMST. Studies investigating differences between GMST and GSAT diagnostics have included physical reasoning (Richardson, 2023) and have assessed the impact of choices of methods used to merge SST and LSAT data (Cowtan et al., 2015; Jones, 2020; Richardson et al., 2018), particularly in regions of changing sea ice cover (Cowtan et al., 2015; Richardson et al., 2018). The IPCC AR6 concluded that "There is high confidence that long-term changes in GMST and GSAT differ by at most 10% in either direction" with "low confidence in the sign of any difference in long term trends" (IPCC WG1 Cross-Chapter Box 2.3) (Gulev et al., 2021). Based on this understanding, and with no available instrumental GSAT data set, the IPCC AR6 derived estimates of GSAT change as equal to the change in observed GMST, with expanded uncertainty ranges to account for differences between the two diagnostics.

The GloSAT reference analysis will extend the available ensemble of estimates of global surface temperature in several ways. Firstly, use of MAT enables an extension of the instrumental record prior to the start of systematic SST observing in the 1850s. Secondly, this earlier extension means that data adjustments are required for the effects of solar heating of the measurements, these have been developed for both marine and land observations and applied in GloSAT. Thirdly, because climate models show different evolution of GMST and GSAT, improving our instrumental records of both measures will help inform ongoing discussions about resolving the disagreements between methods and datasets. And finally, SST and MAT observations contain different biases so producing analyses based on both measures samples another dimension of the structural uncertainty inherent in estimates of global surface temperature. The name "reference analysis" is adopted because the data set is intended to provide a reference for comparison of observed GMST and GSAT, with processing based on the processing workflow of the HadCRUT5 GMST data set.

## 3 Methods for producing the GloSAT dataset

### 3.1 Land surface air temperature data processing

#### 3.1.1 Land station data acquisitions and quality control

The GloSAT reference analysis (GloSATref) uses an updated database of monthly average air temperature at surface meteorological stations. This is an extended version of the CRUTEM5 station database (Osborn et al., 2021), including additional station data, updated methods for climatological normal calculation (Taylor et al., in prep.) and new bias adjustments for early instrumental measurements prior to the adoption of louvred Stevenson-style screens (Wallis et al., 2024).



The new acquisitions and improved/updated series are summarised in Table 2. Quality control is applied to these data at source by the national meteorological services together with further quality control following the methods described in Osborn et al. (2021). These include station neighbour and station extreme threshold checks with threshold modifications based on neighbour stations, and identification and removal of physically implausible values (taking into account month of year, station elevation and station latitude).

### 3.1.2 Early record exposure biases

Measurements in the early record were made in thermometer shelters of widely varying designs. Disparities between measurements made using these different shelters were already being noted in the mid-19th century (Naylor, 2018) and the process of understanding those differences, and the more general difficulties of making reliable and practical measurements across a global network of observers, led eventually to standards for the siting and design of instrument shelters such as those provided by the World Meteorological Organisation (2018, WMO). The transition from non-standard shelters to Stevenson-style screens is a known source of systematic bias – called exposure bias – because earlier shelters were subject to differences in ventilation and direct or indirect radiative heating of the thermometer. Resulting biases exhibit seasonal structure with an identifiable impact on regional trends (Parker, 1994).

Global data sets have partially accounted for exposure biases through their general statistical homogenisation processes and/or by inclusion of a component in their uncertainty models, developed from previous exposure bias assessments (Brohan et al., 2006; Folland et al., 2001; Menne et al., 2018; Morice et al., 2012). Only a limited number of data sources have explicitly adjusted for exposure biases in early temperature records; e.g. in CRUTEM5 only some data sources for Spain (Brunet et al., 2006, 2011), the Greater Alpine Region (Böhm et al., 2009), and Australia (Ashcroft et al., 2014) have had adjustments applied. Specific exposure bias adjustments have not been developed and applied more widely because quantitative estimates of the bias and comprehensive metadata describing screen types in use over time at each meteorological station have been lacking.

Wallis et al. (2024) made a significant step forward in addressing these gaps by compiling temperature observations from 54 parallel measurement series at sites where multiple thermometer shelters were simultaneously in operation and using them to develop regression-based predictive models of the bias between measurements taken in Stevenson-style screens versus those in earlier shelters. These regression models were selected and calibrated using parallel measurements for four classes of shelter:

– Open exposures that, barring shielding to the top and one side, fully expose the thermometer to the air, including Glaisher and Montsouris stands. Bias predictors are annual temperature and a climatology of surface solar radiation.

– Intermediate exposures that provide additional lateral protection to the thermometer, including thermometer sheds and summer houses. This bias prediction model was not applied because its skill arose mostly from a single parallel measurement site.

– Closed exposures in which the thermometer is fully shielded on all sides, including Wild huts. Bias predictor is a climatology of surface solar radiation.





– Wall-mounted exposures, including wall-, fence-, or window-mounted measurements, both screened and unscreened.
       Bias predictor is top of atmosphere downward solar radiation.

Wallis et al. (2024) then compiled a database of shelter types for the majority of mid-latitude stations that had some pre-1961 data in the GloSAT station database. This metadata provides an estimate of the shelter types in use up to the time that a Stevenson-style screen was introduced and allows a prediction of the exposure bias arising from the transition from these

earlier shelters to the Stevenson-style screen using the aforementioned regression models.

The bias estimates from Wallis et al. (2024) have been applied here to adjust the GloSAT station database for exposure bias. Of the 5,031 stations between 30° and 60° latitude (in either hemisphere) with some data prior to 1961, exposure bias was considered to be absent in 1,898 stations (either because they had already been adjusted or because the metadata suggests a Stevenson screen had been in place from the start of the record). Of the remaining 3,133 stations, new exposure bias adjustments

were applied to 1,960 stations. No adjustments were made to 1,173 stations either because shelter metadata were absent or because Wallis et al. (2024) had not found an acceptable bias prediction model (e.g. for intermediate exposures).

While Wallis et al. (2024) provides a useful step forward, residual exposure biases are still present in the GloSAT station database. A sizable fraction of the station records either reside in tropical or high latitude regions that are not assessed by Wallis et al. (2024) or do not have the metadata required to apply adjustments. Additionally, transitions from very early exposures

(such as thermometers placed in unheated poleward-facing rooms) have not been considered. Hence, the ensemble uncertainty model for exposure biases from Morice et al. (2012) is used here, without any reduction, to provide conservative bounds on remaining exposure-related uncertainty in regional and global averages.

### 3.1.3 Improved and additional station normals

The GloSATref gridding process requires the monthly temperatures to be expressed as anomalies by subtracting an estimate of

each station's average (hereafter "station normal") during the 1961-1990 baseline. In CRUTEM5 (Osborn et al., 2021), station normals were computed for each calendar month where at least 15 out of 30 years of data were available in the 1961-1990 period. For stations where this criterion was not met, some station normals were obtained from the WMO (World Meteorological Organisation, 1996) or were estimated by computing 1951-1970 normals which were then adjusted to represent the 1961–1990 mean based on the difference between the grid-box averages for 1961–90 and 1951–70 at nearby locations (Jones

et al., 2012; Jones and Moberg, 2003). Applying this approach for the GloSAT reference analysis would lead to 2,656 stations being unused due to the absence of an estimated normal and some estimated normals would have greater uncertainty and small biases (Calvert, 2024; Taylor et al., in prep.) due to being estimated from incomplete data.

For the GloSAT reference analysis, therefore, the CRUTEM5 approach is revised to augment the available station observations with individual monthly values estimated by the Local Expectation Kriging (LEK) method described by Taylor et al.

(in prep.). The local expectation at a given station location is estimated as a linear weighted average of the temperature values recorded at other stations in the neighbourhood, with the weightings determined by an approximation to Kriging with hold out and taking into account the covariance between stations. The monthly values estimated by LEK are used to fill in missing



values within each station time series, but only for the purpose of calculating station normals from complete 1961-1990 values. Taylor et al. (in prep.) evaluate these calculated normals against normals calculated directly from observed values and find a
root-mean-squared difference of approximately 0.2 °C with no systematic dependence on latitude.

For GloSATref, the normals are therefore calculated from complete 1961-1990 observations (5,806 stations), from a mixed set of observed and LEK-estimated values during 1961-1990 (3,568 stations, with reduced bias for cases where the observed values were either at the start or end of the baseline period), or from only LEK-estimate values (1,742 stations, typically short-segment stations during earlier or very recent periods). This permits use of station series for which few or no observations are
available in the 1961-1990 period and reduces the use of less reliable WMO normals.

In addition to extending the record back to 1781, the total number of stations in the database is increased from 10,639 in CRUTEM.5.0.1.0 to 11,873 in the GloSAT land station database. The number of stations with station normals, and thus able to be used to create the gridded data set, is increased from 7,983 stations in CRUTEM.5.0.1.0 to 11,135 stations for GloSATref.1.0.0.0 (Table 3). Together with new data acquisitions, the use of LEK-derived normals increases the number of
usable stations. Many of the stations with new normal information are, however, situated in relatively well observed locations, and cover relatively short periods of time (mean station length is 35 years for those with normals fully or partially calculated from LEK estimates, versus 94 years for those with full 1961-1990 data), hence the spatial coverage of the globe is not increased commensurately with the increased number of stations with climatological normal estimates.

### 3.2 Marine air temperature data processing

This section describes the processing used to construct a 200-member ensemble of gridded MAT fields that forms the input to the GloSATref spatial analysis. We refer to this MAT dataset as GloSATMAT. The production of stable GSAT climate records requires assessment of the systematic biases and uncertainty in MAT observations. The most important of these are diurnal biases and measurement height changes.

Diurnal effects are caused by solar heating of the ship which in turn warms the air around the sensor and have been addressed
previously through two approaches. The first, and most common, approach is to discard all observations affected by daytime heating to produce a nighttime marine air temperature (NMAT) data set (Bottomley et al., 1990; Cornes et al., 2020; Junod and Christy, 2020; Kent et al., 2013; Rayner et al., 2003), with the definition of nighttime typically being from one hour after sunset to one hour after sunrise. Although simple to apply, this restricts the starting date of NMAT datasets to the late nineteenth century as before that time an increasing proportion of observations were taken during the daytime and commonly at
local noon (Kent and Kennedy, 2021). The second approach is to model and adjust for daytime biases. This has the advantage of increasing the sample size of observations and was the approach taken by Berry and Kent (2011) who produced a global gridded all-hours air temperature product starting in the 1970s. This approach is also taken here but is used to construct a MAT dataset back to 1784. Adjustments from the measurement height to a standard reference height are required as the temperature typically decreases with increasing height above the ocean as the sea is usually warmer than the air above. Height adjustments
have been made by estimating the lapse rate of the lower atmospheric boundary and using this information with known values (or estimates) of the temperature recording height to adjust the temperature to a common reference height.





Processing of GloSATMAT largely follows Cornes et al. (2020) with the following description of the method focusing on additions and variations from that approach. The most notable changes are:

- A refined Quality Control (QC) procedure

- Development of an ensemble version of the Cornes et al. (2020) error model to produce a 200-member ensemble data set

- The inclusion of observations of marine air temperature made during the daytime, following adjustment for daytime heating bias (Cropper et al., 2023) using the method proposed by Berry et al. (2004)

A new version of CLASSnmat (version 2.1.0.2) has also been produced (Cornes et al., 2024), which uses the same input
data and methodology as GloSATMAT but is constructed from nighttime-only values and hence omits the diurnal adjustments. CLASSnmat v2 only extends back to 1880 due to the reduced sampling of nighttime observations before that time.

### 3.2.1 Marine observations and quality control

The principal source of ship data is the International Comprehensive Ocean-Atmosphere Data Set (ICOADS, Freeman et al.,
2018; Li et al., 2021) and data are processed from 1784 to present using the methods described in Cornes et al. (2020). In addition, ship data from the following sources are also included:

- Citizen science data digitisation undertaken under the Old Weather initiative (Spencer et al., 2019)

- Research Vessel data from ICOADS holdings of the Ship-based Automated Meteorological and Oceanographic System (SAMOS) archive (Smith et al., 2018)

- Voluntary Observing Ship Global Data Assembly Centre (VOS-GDAC), obtained on 19th January 2021 https://www. dwd.de/EN/ourservices/gcc/gcc.html. In cases where an observation in ICOADS has the same ID, date, and position as a VOS-GDAC observation, the information from the VOS-GDAC observation is used.

As in Kent et al. (2013) and Cornes et al. (2020), data for the period 1876–1893 for ships passing through the Suez canal were excluded due to the warm bias in those observations.
The QC checks applied to the MAT data follow those described in Cornes et al. (2020). An updated climatology-based quality control check has been developed for the MAT data, described below. For GloSATMAT additional checks have been made in relation to the diurnal heating bias adjustments (Cropper et al., 2023).

*Climatology QC procedure*

At the start of marine data processing climatological outliers are rejected based on the climatology generation step. Under
the new procedure, a 1961-1990 pentad (5-day) climatology is generated on a 1° latitude by 1° longitude resolution grid. Due to the sampling frequency of ship-based observations, if a grid cell does not initially contain at least once value in each pentad





throughout the year or one value in each month, the size of the grid cell, centred on the target 1° by 1° grid cell, is expanded in 1° latitude and longitude increments. This occurs until either the criterion of 500 values across at least 48 pentads and 10 months is met or there is one complete set of pentads. These criteria prevent climatologies being formed from data with preferential

sampling in one part of the year. Harmonics are fitted to the pentad averages per grid-cell and an optimum number of functions is chosen using the Akaike Information Criterion up to a maximum of three harmonics. The fitted harmonic coefficients are then used to generate a daily climatology for each grid cell.

Using the grid-cell climatology values, outliers are identified from the distribution of anomaly values per year. Values that exceed a lower or upper bound from the distribution are flagged for exclusion. These bounds, $b_{\mathrm{lower}}$ and $b_{\mathrm{upper}}$, are defined by

the 5th and 95th percentiles ($p_5$ and $p_{95}$), the 5th to 95th percentile range and an expansion factor ($f_{\mathrm{exp}}$).

$$b_{\mathrm{lower}} = p_5 - 0.5 f_{\mathrm{exp}}(p_{95} - p_5) \tag{1}$$

$$b_{\mathrm{upper}} = p_{95} + 0.5 f_{\mathrm{exp}}(p_{95} - p_5) \tag{2}$$

To exclude gross outliers, an initial check is performed across each latitude band per year with $f_{\mathrm{exp}} = 2$. Following exclusion of values that fail that check, the outlier flag is applied to observations with $f_{\mathrm{exp}} = 0.5$ on the highest grid cell spatial resolution

available (starting at 1° resolution) that contains 50 observations per month, progressively increasing the grid box size in 1° increments. If, despite that increasing box size, the number of observations is still less than 50 then the distribution is formed from anomalies across the 10° latitude band per month.

The generation of the climatology and the quantile-based removal of values are applied as follows. A climatology is generated using data that passes the Met Office QC checks (after Cornes et al., 2020). The above quantile-based QC procedure is

then applied and then a new climatology is generated using data that passed the first quantile-based QC check and a second iteration of quantile-based QC is applied.

*Diurnal bias-related QC procedure*

Additional QC is applied as part of the diurnal bias adjustment process (see section 3.2.4) as documented in the appendix of Cropper et al. (2023). Four modifications to the QC method described in Cropper et al. (2023) are made here:

– precipitation-flagged observations were retained for use in the analysis (but excluded from the heating bias fitting process)

– all ships with unrealistic diurnal heating were excluded, not just those in the 1854-1894 period described in Cropper et al. (2023)

– observations with an estimated heating bias $\geq 15$ °C were also flagged and excluded from the analysis.

– for ships lacking ID information only nighttime observations are retained as IDs are required for diurnal bias adjustment





### 3.2.2   MAT measurement height adjustments

In HadNMAT2 (Kent et al., 2013) and CLASSnmat v1 (Cornes et al., 2020) the measurement height uncertainty is represented as a standard deviation around the estimated or known measurement height. This uncertainty is propagated through the height adjustment calculation (see following section) using a Monte Carlo approach sampling a distribution of heights and joint distribution of wind speed and air-sea temperature difference to estimate the likely uncertainty in atmospheric stability. Details can be found in (Kent et al., 2013) with updates in (Cornes et al., 2020).

*Adjustment of MAT observation to a standard height above sea level*

The previous approach provides only a likely distribution of measurement height and height adjustment uncertainty. Correlated and uncorrelated contributions were estimated separately, and the correlation structure was not captured. For those observations that could not be linked to a measurement height in WMO Publication 47 (Kent et al., 2007), default heights were chosen which after 1945 accounted for the regional variation in typical ship heights. However, this had the consequence that a ship with a known ID and unknown height would change its estimated height depending on which 5° grid cell it occupied at a particular time. Here, the development of an ensemble of heights permits correlated uncertainty to be handled as follows:

– In each ensemble member a ship with known ID will keep the same measurement height, at least within a calendar year

– In each ensemble member the stability estimate will be the same for nearby ships within the same 5° grid cell and 10-day period.

Prior to height information being available for individual observations from WMO Publication 47 (mostly pre-1973 before callsigns are available to assign WMO Publication 47 entries to ships in ICOADS), observation heights are estimated from the literature following Kent et al. (2013). A 200-member ensemble is created to reflect uncertainty in measurement heights; the parameters for the ensemble are shown in Table 4. Per ship, the ensemble varies the height four times to reflect the change in average ship height over time (parameter prefix "Height" in Table 4). The exact timing of the start of the height changes also forms a component of the ensemble to reflect uncertainty in that date (parameter prefix "Year" in Table 4).

From 1970 onwards, the height of a ship is determined by joining the ICOADS ID/callsign with the corresponding WMO Publication 47 entry. Sometimes, the height of the dry bulb thermometer above the summer load line i.e. thermometer height, is used. However, if the thermometer height is missing, then the height of the barometer, anemometer or the height of the visual observing height is used. If the height of the ship's thermometer (inferred from WMO Publication 47) is known, the assumed uncertainty in that height is ±1 m.

For missing heights after 1970, the height is sampled from the distribution of known heights from WMO Publication 47. Height information is grouped by 5° grid box, 10° grid box, country of registration, oceanic basin, 10° latitude band, vessel length and vessel type. For IDs without a height, the height is sampled across a selection of heights where a ship has matching information. Observations without ship ID information are assigned a pseudo-ID and placed in a sub-group for which an ensemble of possible heights is generated. The overall estimated thermometer height for ships in ICOADS is shown in Fig. 2 of Cornes et al. (2020).



To adjust MAT measurements to a standard reference height, and quantify uncertainty in adjustments, we implement an
290 ensemble version of the height adjustment method described in Cornes et al. (2020). For each ensemble member, a single
random sample of stability parameters (temperature scaling parameter and Monin-Obukov length, Biri et al., 2023) is taken
in each 5° grid cell and 10-day period from a pool of 10,000 parameter sets, which comprises 5,000 samples taken from the
distribution of random uncertainty components and 5,000 taken from the systematic components (see Cornes et al., 2020). A
fixed sample over each 10-day period within a month is used. This is designed to replicate the expected constancy of stability
parameters within the synoptic timescale. For each sample of stability parameters and height estimates, the temperature data
were adjusted to a height of 2 m above sea-level.

### 3.2.3 Adjustment of observations during World War 2

The adjustment of data from ICOADS Decks 245 (UK Royal Navy Ships) and 195 (U.S. Navy Ship Logs) during the period
1942-1945 follows the method described in Cornes et al. (2020) to calibrate observations from these decks to those in other
decks. For GloSATMAT the adjustments and uncertainty values have been calculated for both daytime and nighttime data.
While the uncertainty in these adjustments was previously considered to be entirely correlated by ship track, the World War
2 adjustments have been partitioned so that 1/3 of the total uncertainty is modelled as correlated by ship track and 2/3 is
uncorrelated. This partitioning addresses a problem with the error covariance matrices, which would otherwise have not always
been positive semidefinite and hence could not have been used in further calculations.

### 305 3.2.4 Diurnal heating bias adjustment

The Cropper et al. (2023) implementation of the Berry et al. (2004) heating bias model was used to quantify the expected
diurnal influence on MAT of energy storage and release by ship superstructures. On an annual basis, we fit a set of heating
bias model coefficients to every uniquely identifiable ship track used from ICOADS following processing to improve the
association of ship identifiers with individual ships as described in Cornes et al. (2020). In addition, some ship IDs prior to
1850 were not unique and were assigned new unique IDs using additional information contained within the ship report. Where
this was possible the data for these problematic IDs were retained, otherwise the data were excluded. The estimates of the
heating bias are based on the difference from MAT and the underlying trend in nighttime temperature, defined as temperature
between one hour after sunset until one hour after sunrise. Hence, each MAT observation is adjusted by the difference between
the estimated bias and the nighttime mean MAT. As such, the full diurnal cycle is removed from the data and GloSATMAT
should be considered a nighttime-equivalent dataset.

As described in Cropper et al. (2023), 2500 alternate sets of heating bias model coefficients are determined from the full suite
of ICOADS ships from 1856-2020. For each individual ship, 2500 realisations of the heating model coefficients are generated,
and an ensemble of 60 of these, minimising several different cost functions, are selected as the best fit for each ship. The
ensemble means of the adjustments over the 60 different ensemble members becomes the heating bias adjustment.

The data requirements for each observation to apply the heating bias adjustment are position, time, cloud cover and relative
wind speed. For a ship track to be included, we require at least 12 observations where the underlying nighttime MAT can be

determined. If a ship track has partially missing cloud and/or wind speed values, these values are sampled from the 1961-1990 climatological distribution of cloud and wind derived from ICOADS after the 60 best model coefficient combinations have been selected. If a ship track has completely missing cloud and/or wind, the sampling occurs during the selection of the heating

bias model coefficients.

Almost all observations pre-1856 lack cloud cover data and many ships do not meet the criterion of having 12 observations for which the underlying nighttime MAT can be determined. In this period, we relax this requirement so that all observations in this period that pass QC can be included. For many ships this means that the coefficients for the heating bias model cannot be fitted as there is no target nighttime information. In this situation we use an ensemble of heating bias model coefficients

computed from ships that do have the required data, where it is likely that these ships are typical during this period. The correlated uncertainty for MAT from ships fit this way is fixed at 0.45°C, the 97.5th quantile over the 1857 - 1870 period.

### 3.3 The GloSAT GSAT reference analysis

#### 3.3.1 LSAT Anomaly Grids

A 200-member ensemble of gridded LSAT fields has been constructed from station temperature series by applying the ensem-

ble gridding procedure described in Morice et al. (2012). This includes ensemble sampling for homogenization uncertainty, exposure bias uncertainty, and urbanization uncertainty, which are accompanied by analytical estimates of measurement and sampling uncertainties as in Morice et al. (2012). The exposure bias model of Morice et al. (2012) is retained despite the addition of new exposure bias adjustments as the Wallis et al. (2024) adjustments are available for a subset of stations located at mid-latitudes only and the adjustments are uncertain. Table 5 shows the mapping of error model terms to the uncertainty

model's ensemble members and error covariance matrices. These ensemble members and error covariance matrices form the inputs for a non-interpolated merged analysis and the resulting 200-member LSAT anomaly ensemble grids are then used as inputs to the Gaussian process based infilling method described in Morice et al. (2021) and previously used to create the HadCRUT5 dataset (see section 3.3.3).

#### 3.3.2 MAT Anomaly Grids

A 200-member marine air temperature ensemble has been generated based on the 200-member ensemble of diurnally adjusted all hours marine air temperature observations adjusted to a 2 m reference height. Each ensemble member is gridded separately after the method described in Cornes et al. (2020) using the ship observation ensembles that sample uncertainty in height adjustments and World War 2 (WW2) adjustments. As in CLASSnmat v1, these gridded fields are initially generated as monthly actual MAT values and then climatological averages are subtracted from the gridded data to produce the anomaly fields. Missing

values in the GloSATMAT climatology fields are filled using a thin-plate spline with the smoothing parameter set to zero. In this way, the spline acts as an exact interpolator, and hence where grid-cell values exist these are reproduced exactly in the interpolated field. Missing cells are interpolated from non-missing cells in the neighbourhood, and this allows anomaly values to be calculated in regions where there are insufficient values over the climatological period to construct a 30-year mean value.





The gridded anomaly values are calculated by subtracting this climatology from the gridded actual temperature fields for the
respective month of the year. As a final QC check on the gridded data, any grid-cell MAT anomalies greater than 10 °C or
less than -10 °C are removed. It should be noted that these climatologies differ slightly from the values used in the earlier
climatological QC (see section 3.2.1) of the marine data due to that being a different stage of the processing chain.

Following the example of the CLASSnmat dataset, uncorrelated, systematic, and sampling uncertainties are encoded into
monthly error covariance matrices, excluding terms included in the ensemble. Uncertainty in diurnal heating adjustments is
encoded into these error covariance matrices and treated as uncorrelated between observations. A summary of uncertainty
model components is provided in Table 5. The 200-member gridded MAT ensemble and accompanying error covariance
matrices are provided as inputs into the Gaussian process based infilling method (previously used to create the HadCRUT5
dataset, Morice et al., 2021) to generate a 200-member infilled MAT anomaly ensemble.

### 3.3.3  LSAT and MAT interpolated analyses

Interpolated analyses of LSAT and MAT are produced using the Gaussian process analysis system of Morice et al. (2012).
This analysis method takes ensemble gridded temperature anomaly fields as inputs, modelling systematic error structures in
observed air temperature anomalies. Additional observational error structure is provided to the analysis in error covariance
matrices. Full details of the method are provided in Morice et al. (2021).

Separate LSAT and MAT analyses are produced. Ensemble and error covariance inputs to the analysis equations are sum-
marised in Table 5. The LSAT analysis error model structure matches that used in the HadCRUT5 data set, with the structure
of individual uncertainty components described in Morice et al. (2021). The structure of terms in the MAT, encoding error
structure into the 200-member MAT analysis ensemble and accompanying error covariance matrices, are described in Section
3.2 and in Cornes et al. (2020) and Cropper et al. (2023).

Analysis estimation follows the method of Morice et al. (2021) using the HadCRUT5 analysis processing system. As in
Morice et al. (2021), the Gaussian process models monthly temperature anomaly fields as the sum of a monthly field mean and
a spatially varying Gaussian process. The Gaussian process model uses a Matérn covariance function that models covariances
between locations on the Earth's surface as a function of Euclidian distance. As for HadCRUT5, the Matérn covariance func-
tion's smoothing parameter is set to $\eta = 1.5$. Parameters representing the standard deviation of temperature anomaly variability
$\sigma$ and spatial decorrelation length scales $\rho$ are estimated separately for land and ocean analyses as the average of maximum
likelihood estimates for monthly fields from 1961 to 1990. The resulting parameter estimates for the LSAT analysis are $\sigma = 1.2$
°C, $\rho = 1300$ km and for the MAT analysis are $\sigma = 0.65$ °C, $\rho = 1550$ km. For comparison, these LSAT parameters estimates
for GloSATref are the same as for HadCRUT5, but the GloSATref MAT parameter estimates differ from those for SST in
HadCRUT5 which have a lower amplitude ($\sigma = 0.6$ °C) and shorter length scale ($\rho = 1300$ km).

As in the HadCRUT5 data set, the analyses are masked in regions of weak observational constraint, defined by a metric of
one minus the ratio of posterior to prior variance of the spatial model estimates at each analysis grid cell. The analysis is masked
where this metric takes a value less than $\alpha = 0.25$ (see Morice et al. (2021) and its supporting information for discussion).



### 3.3.4 Merging LSAT and MAT analyses

The GSAT ensemble grids are produced as weighted averages of MAT and LSAT anomaly ensemble grids. The weighting scheme follows the HadCRUT5 method, with weighting based on the fraction of land area in each grid cell. As in HadCRUT5, for the interpolated analysis, sea ice regions are treated as if they were land in the weighting scheme and a minimum weighting of 25% land is placed on grid cells that are directly observed by land meteorological stations. These choices are retained from HadCRUT5 to aid comparison of the climate diagnostics based on SST and MAT through comparison of HadCRUT5 and the GloSAT reference analysis.

## 4 Results and Discussion

### 4.1 Global and hemispheric temperature anomaly series

Figure 1 shows a comparison between GloSATref and HadCRUT5 interpolated global and hemispheric analyses including uncertainty ranges and percentages of global and hemispheric coverage. Each of these two analyses is produced using the HadCRUT5 analysis methodology. Differences in global annual average time series between these two analyses reflect the differences in input LSAT and MAT/SST data sets and their uncertainties combined with the interaction between anomaly patterns and data coverage in those input data sets and the spatial interpolation methods.

GloSATref, like HadCRUT5, represents the major changes we expect to see in the global mean. The pre-1850 record, despite its increased uncertainty, captures the strong cooling associated with major volcanic eruptions. The overall warming of global temperature is clear, although GloSATref warms slightly less overall. Features noted in previous comparisons of SST and NMAT large-scale averages remain prominent in the GloSATref comparison to HadCRUT5, for example the lower temperatures in the early 1900s and in recent years (Cornes et al., 2020; Gulev et al., 2021; Chan et al., 2024).

For their period of overlap, currently 1850 to 2021, there is broad agreement in the global and hemispheric means. While areal coverage of the northern hemisphere is similar in the two data sets, GloSATref has slightly reduced southern hemispheric coverage. GloSATref is on average warmer than HadCRUT5 from the start of the latter dataset in 1850 until 1880 and then more notably cooler on average than HadCRUT5 until the early 1910s, especially in the Southern Hemisphere.

Variability in GSAT and hemispheric averages from GloSATref is high in the period before the start of HadCRUT5 in 1850, as expected from the lower data coverage. There are two periods of strong cold anomalies which result from the cooling of the atmosphere following the eruptions of major volcanos: Laki in 1783-1784 (Zambri et al., 2019), an eruption of unknown source in 1808, Tambora in 1815, and further eruptions in the 1820s and 1830s (Brönnimann et al., 2019). It is likely that the global mean effect of Laki is over-estimated in this time series (Figure 1a) because it is influenced by the location of the available observations in the 1780s; this effect can be seen in model simulations masked to the GloSATref data coverage (Ballinger et al. in prep.).

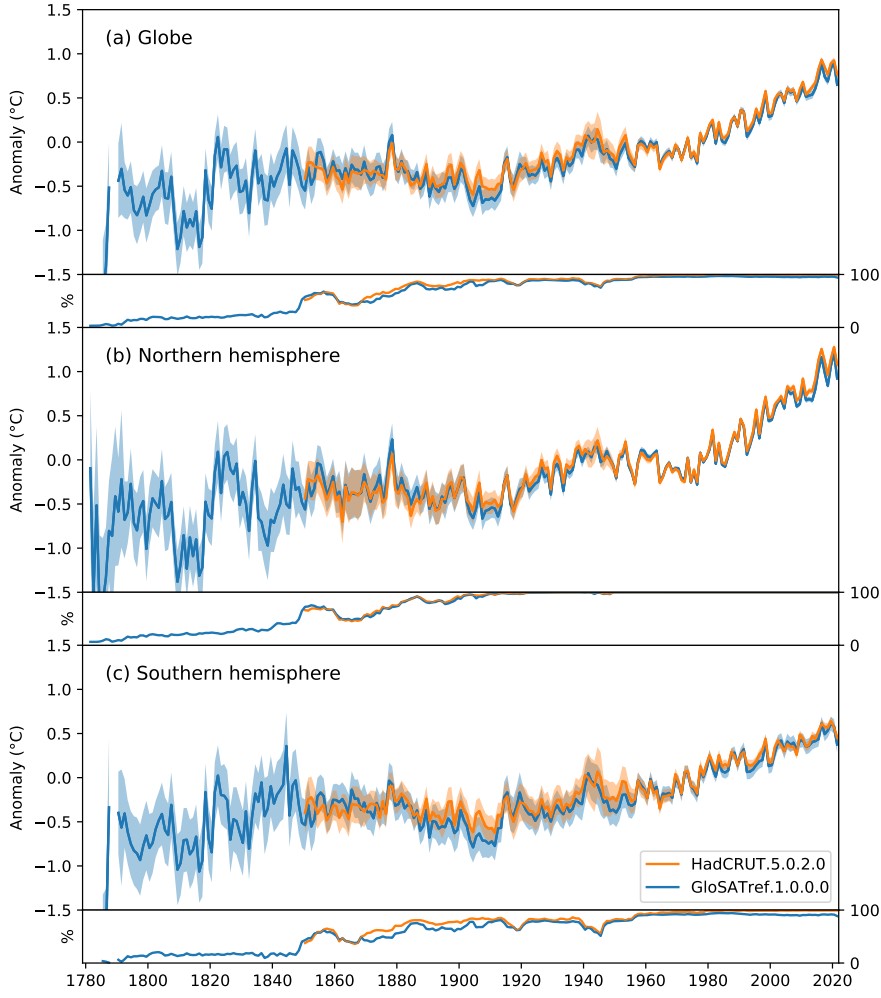

**Figure 1.** Global and hemispheric average temperature anomaly time series for the GloSATref and HadCRUT5 global analyses (°C, relative to 1961-1990), together with the % of areal grid coverage for (a) the full globe (b) the northern hemisphere and (c) the southern hemisphere. Hemispheric series omit years in which there are no data available in the respective hemisphere. Global averages are computed as an average of northern and southern hemispheric series, requiring data to be available in both hemispheres. Methods for computation of time series and their uncertainties are provided in (Morice et al., 2021).

## 4.2 Temperature anomaly maps and data coverage

The most obvious feature in the 20-year averages of SAT anomalies shown in Fig. 2 is the increase in temperature over time,
particularly at high northern latitudes. The period before about 1820 is particularly cold, likely due to volcanic activity. Warm anomalies are present in higher latitudes of the North Atlantic in the 20-year averages from 1821-1840 to 1900 and the South Atlantic from 1821-1840 to 1861-1880. These might be indicative of multidecadal variability. The Arctic region is relatively


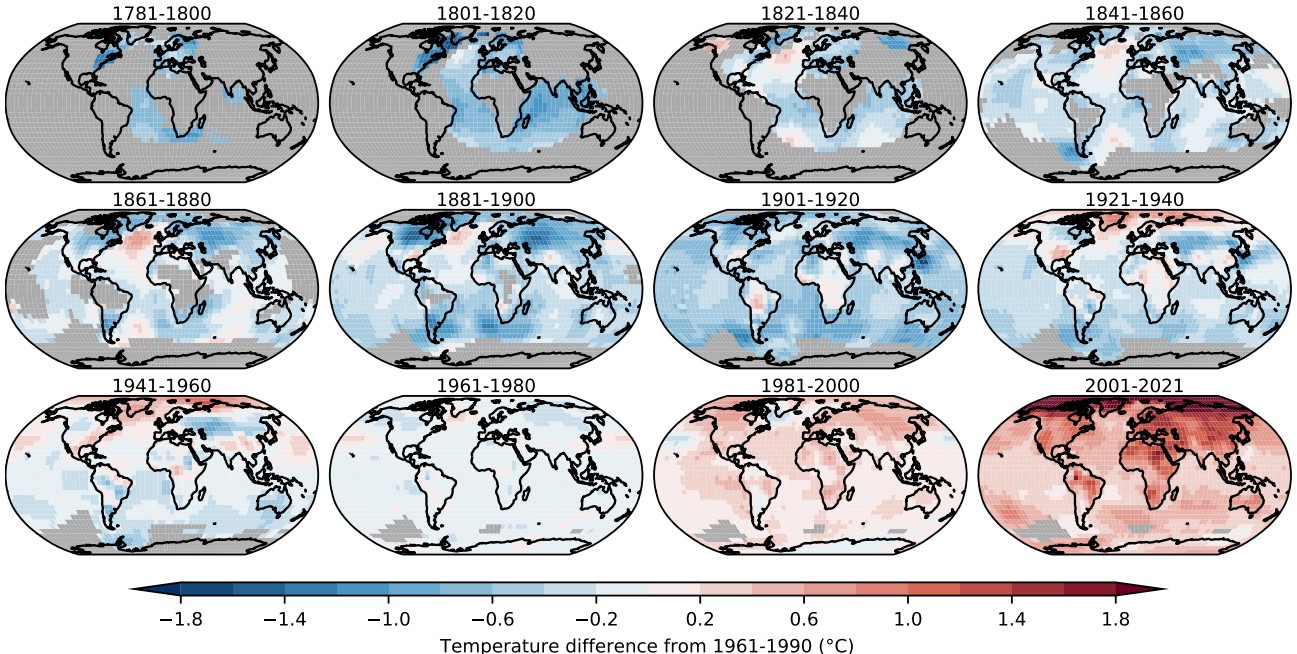

**Figure 2.** 20-year average SAT for the GloSATref analysis (°C relative to 1961-1990, final panel shows 21-year average). Each panel averages available gridded data to quarterly, annual and then 20-year (21-year) averages, requiring data in at least two quarters to form an annual average and at least 10 annual averages to produce a 20-year average.

warm in the 1920s and 1930s (Hegerl et al., 2018). Despite the new adjustments to MAT for biases present during WW2, this period may still be unrealistically warm in GloSATref according to a new analysis of SST data (Chan et al., 2024).

While use of MAT allows GloSATref to extend prior to 1850, there is nevertheless a marked reduction in data availability prior to about 1855 (see global coverage metrics in Fig. 1 and 20-yearly coverage maps in Supplement Fig. S1 and Fig. S3). Before 1850, the majority of weather station LSAT series are situated in Europe and the eastern coast of North America, limiting northern hemisphere coverage of the GloSATref analysis fields. Combined with an absence of station records in the southern hemisphere, this contributes to increased uncertainty in global and hemispheric series. MAT coverage is predomi-

nantly restricted to the Atlantic and Indian Oceans, reflecting primary trade routes. For these regions it is possible to make estimates of regional temperature anomalies based on GloSATref analysis fields using reasonable criteria for data availability (Fig. 2). The Pacific has highly limited sampling in this early period. Non-uniform global coverage in the early record may lead to a sampling bias in global average SAT estimates in this period. Uncertainties in global and hemispheric means are therefore larger in this early period as both LSAT and MAT are sparsely observed, but the uncertainty estimates themselves are also

likely to be more uncertain.

Observation coverage improves over time accompanied by increased global coverage of the gridded analysis fields. In the second half of the 1800s there is a marked increase in station LSAT series availability, with analysis estimates available by the



end of the century for most land regions excluding Antarctica, interior regions of Africa, northern regions of South America including the Amazon. By the 1880s measurements from all oceans are available, although with varying degrees of data availability. While sufficient data is available to estimate average temperature anomalies across much of the Pacific at this time (Fig. 2), data coverage in the analysis is limited in regions of the western and southern Pacific. Data coverage for the Southern Ocean and high northern latitudes is highly limited.

In the early 20th century most of the missing grid cells are in the Southern Ocean and over Antarctica. Underpinning land station data coverage remains limited for much of Africa, South America, interior regions of Eastern Asia and high latitude regions of North America and Eurasia, although the available observation coverage permits analysis estimates for these regions. The Antarctic becomes represented in the analysis in the 1950s. Coverage for MAT peaks in the 1970s and 1980s and has declined markedly since (Berry and Kent, 2016; Kent and Kennedy, 2021). Observation coverage of the Southern Ocean and southern extents of the Pacific, Atlantic and Indian Ocean remains reduced in GloSATref in comparison to HadCRUT5 in recent decades (see Supplement Fig. S1-S4). SST observations from drifting buoys contribute notably to differences in marine data coverage in these regions but there are also fewer ship observations of MAT than of SST.

As in the preceding CRUTEM5 station database (see discussion in Osborn et al., 2021), the number of LSAT series is relatively high and stable from the early 1960s to the early 2010s, with a sharp reduction in the last decade due in part to delays in access to data. However, in CRUTEM5, the number of station series actually used in the gridding showed a decline from the 1970s onwards, falling to below 90% of its 1970s peak from 1991 onwards (figure 7 of Osborn et al., 2021) and this is mostly because new stations installed since the 1970s did not have the data necessary to estimate their 1961-1990 normals. In the current study, the use of kriging (section 3.1.3) to estimate missing values from neighbouring stations for the purpose of estimating normals has partly ameliorated this issue: the number of stations used for gridding is 90% of its 1970s peak as late as 2011 (compared with 1991 in CRUTEM5). It is worth noting, though, that these additional stations that can now be used for the gridded dataset are commonly in grid cells with other LSAT stations; therefore, their importance is that they reduce the sampling uncertainty at the grid cell level rather than extending coverage of the gridded dataset in the final decades.

Differences in temperature anomalies over land between GloSATref and HadCRUT5 (Fig. 3) result from the increased number of stations used to create the grids, primarily from use of the LEK method, and from early-record mid-latitude screen bias adjustments. The effects of screen bias adjustments are evident in anomaly difference maps prior to the 1930s, most notably with GloSATref containing cooler anomalies across Eurasia in the 1890-1909 and 1910-1929 panels and in western North America in the 1850-1869 panel. This difference is largest in summer and autumn months when northern hemisphere screen biases are largest, for which HadCRUT5 does not include adjustments. From the 1930s onwards, differences between GloSATref and HadCRUT5 anomalies over land are minimal.

## 4.3 Effects of MAT and LSAT updates in global series

Averaging globally over the land domain for the non-interpolated GloSATLAT data (noting that coverage becomes increasingly limited during the first 100 years) and comparing to the closely related CRUTEM5 data set (Figure 4a,d) shows that the Wallis et al. (2024) exposure bias adjustments cool LSAT global annual averages in the 19th and early 20th centuries by less than 0.1

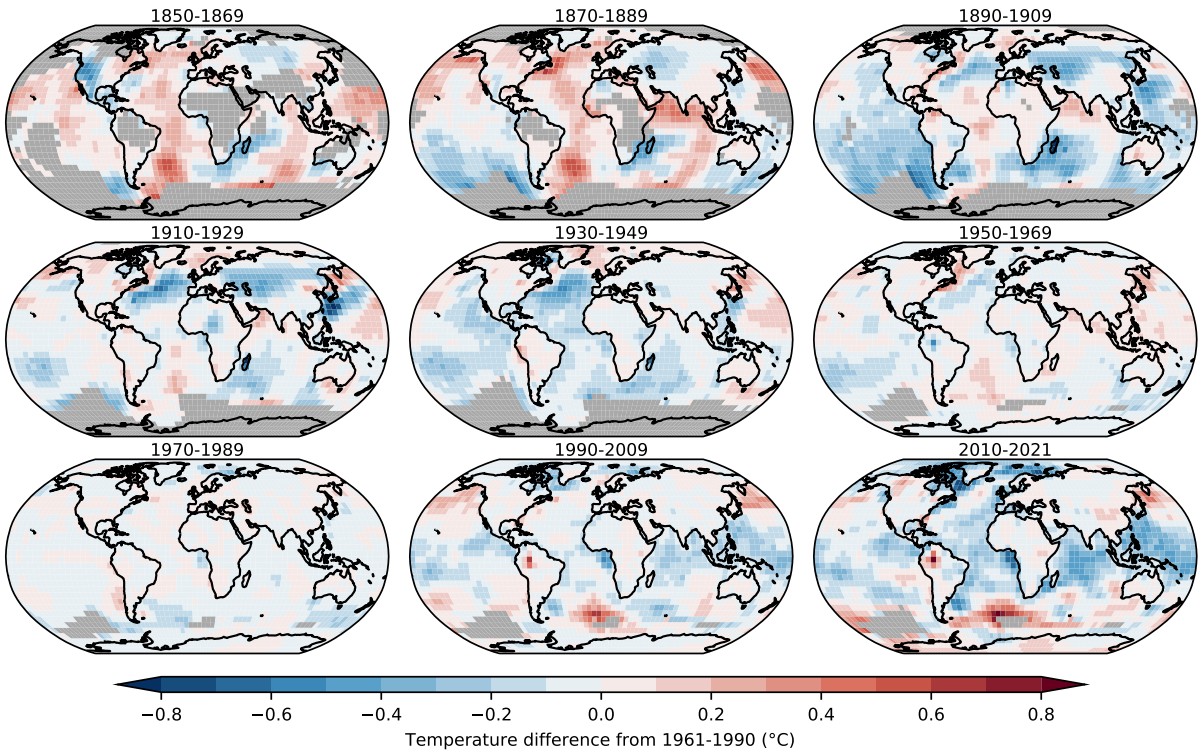

**Figure 3.** Difference in 20-year average anomalies, relative to 1961-1990, between the GloSATref and HadCRUT5 analyses (°C, final panel shows 12-year average). Blue colours indicate where GloSATref is colder than HadCRUT5. Red indicates where GloSATref is warmer.

°C, primarily through mitigation of summer warm biases which are larger than the annual bias adjustments. Differences from CRUTEM5 in the 21st century are less than 0.04 °C of either sign. These differences result from the expanded set of station records included in anomaly grids.

Comparing the all-hours GloSATMAT to related NMAT data sets, CLASSnmat v2 uses the same methodology and input data sources as GloSATMAT but, like its predecessor CLASSnmat v1, excludes daytime observations and starts in 1880. The main reason for differences between the two versions of CLASSnmat is the addition of extra data sources (see Sect. 3.2.1, Fig 4b,e). CLASSnmat v2 is therefore most similar to GloSATMAT. For most of the period 1850-1879 where there are only estimates from GloSATMAT and HadSST4, GloSATMAT is warmer than HadSST4 on average by about 0.1˚C, which may

suggest either an incomplete removal of diurnal heating biases in GloSATMAT or a residual cold bias in HadSST4. This relative warmth compared to HadSST4 is also seen for the first decade of the NMAT records. The early 20th century shows rapid variations in the difference between the MAT datasets and HadSST4. Recent analysis has suggested that most SST data products, including HadSST4, may be biased cold during the early 20th century (Sippel et al., in press; Chan et al., 2024); further work is required to understand the different temperature records in this period. During WW2, GloSATMAT shows

cold temperature anomalies relative to both HadSST4 and the NMAT-based estimates. This may suggest that the new bias

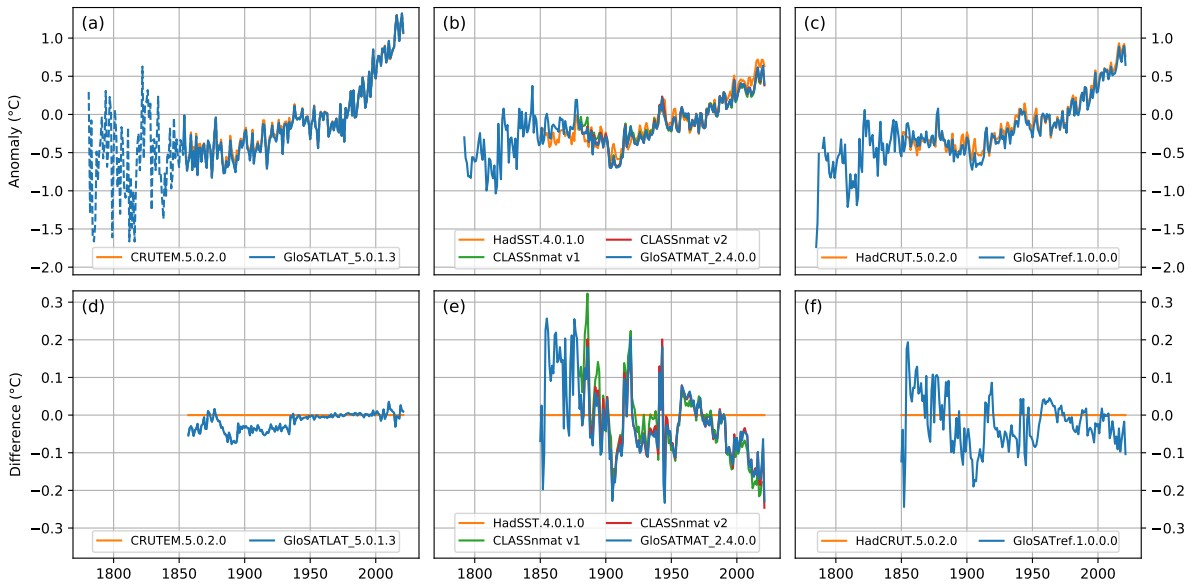

**Figure 4.** Top row: global average temperature anomalies (°C) relative to 1961-1990 for (a) LSAT, (b) SST/NMAT/MAT, and (c) GM-ST/GSAT. Bottom row: differences in global average 1961-1990 temperature anomaly time series for (d) LSAT minus CRUTEM.5.0.2.0 LSAT, (e) SST/NMAT/MAT minus HadSST.4.0.1.0 SST, and (f) GMST/GSAT minus HadCRUT.5.0.2.0. The dashed line for GloSATLAT extends the series prior to 1856 by omitting the requirement for at least 5 populated grid cells in each hemisphere, with only Northern Hemisphere station data present in GloSATLAT grids prior 1854. None of the LSAT/SST/NMAT/MAT data sets in panels (a, b, d, e) use spatial interpolation while spatial interpolation is used in GSAT/GMST analyses shown in panels (c, f). Plotted series use all available gridded data without co-location to common data coverage.

adjustments are an improvement (Sect. 3.2.3) as a new analysis by Chan et al. (2024) indicates HadSST4 may contain residual warm biases during WW2.

The smaller warming trend in NMAT relative to SST data has been noted for some time (for an overview see Cornes et al., 2021; Gulev et al., 2021). This feature is also apparent in the MAT data presented here (Fig. 4) and is shown to start in the late 1950s. SST and air temperature anomaly differences in this period are larger in tropical regions but are more similar during El Nino periods (not shown). Climate models show the opposite relationship between SST and MAT trends (Gulev et al., 2021) supported by a proposed physical constraint (Richardson, 2023). Further analysis of the in situ marine observations is therefore required.

Global average temperature series for GloSATref and HadCRUT5 analyses are shown in Fig. 4c (as Fig. 1 but without the uncertainty range), and their difference in Fig. 4f. Over their common period, comparison with averages derived from their underpinning single domain data sets indicates that differences in time variation of global temperature anomaly series for GloSATref and HadCRUT5 primarily arise from their marine components, with early record LSAT exposure bias adjustments





having a smaller effect.

## 4.4 Comparison to other GMST data sets

The global average temperature time series for the GloSAT GSAT analysis is shown in Fig. 5 alongside those of current GMST analyses. Fig. 5b gives a more detailed view of their differences from GloSATref. As noted in Section 2, the GMST data sets differ in methods used and underpinning observation data although there are overlaps and commonalities between them in both respects.

Also shown is the global average temperature anomaly time series for the median of the PAGES2k multi-proxy ensemble reconstructions (PAGES2k, 2019). The PAGES2k ensemble was calibrated using the Cowtan and Way (2014) spatially infilled analysis of HadCRUT4 data over 1850–2000, so that (by weighting and scaling the proxy records) it resembles the overall warming and some of the variability of the infilled HadCRUT4. The pre-1850 period is not directly constrained in this way and it is for comparison with GloSATref during this period that it has been included.

There is general agreement between all the estimates, with particularly good agreement on the representation of interannual variability among the instrumental-based datasets. From 1850 to around 1885, there is a wider scatter between the various datasets, with some warmer and some cooler than GloSATref reflecting the greater uncertainty in these early observational datasets. This is also a period when these other datasets lie outside the GloSATref 95% confidence range more often, especially the recently-published DCENT dataset which has new treatments of inhomogeneities (Chan et al., 2024). From 1888 to 1913, GloSATref is consistently cooler than the SST-based datasets and they all lie outside the GloSATref 95% confidence range at times during this period except for the HadCRUT5-based Calvert (2024) dataset. This difference is notable because there is evidence that this early 20th century cool period is biased cold in the SST datasets (Sippel et al., in press), yet GloSATref is cooler still by 0.1 to 0.2 °C. The SST-based datasets are more similar to each other than to GloSATref due to differences between SST and MAT (see Fig. 4e).

From the 1930s to the end of the 20th century, differences between GloSATref and GMST data sets based on ERSST v5 (GISTEMP and NOAAGlobalTemp) are on average smaller than differences with HadSST4 based data sets (HadCRUT5 and Berkeley Earth). This is likely related to the more direct use of NMAT observations to bias adjust ERSST v5 SSTs than HadSST4 SSTs. From the 1950s onwards the dominant difference is the slightly lower trend in MAT relative to SST noted earlier. This leads to GMST data sets lying above the GloSATref confidence range in recent years.

Before 1850, the only comparison is with data from the palaeoclimate record. During this period, GloSAT shows much greater variability than it does after 1850. Part of this enhanced variability is likely due to strong volcanic activity, with the GloSATref series cooling around the times of the unidentified 1809 volcanic eruption and the 1815 eruption of Mount Tambora, and to a lesser degree around the time of the 1830s eruptions. However, GloSATref shows greater variability than the PAGES2k median, which extends both above the 97.5% confidence limit and below the 2.5% confidence limit of GloSATref (Fig. 4 and Supplement Fig. S5). There are uncertainties in the representation of variability in the PAGES2k median (PAGES2k ensemble range is not shown here, PAGES2k, 2019; Anchukaitis and Smerdon, 2022). However, it is also likely that pre-1850

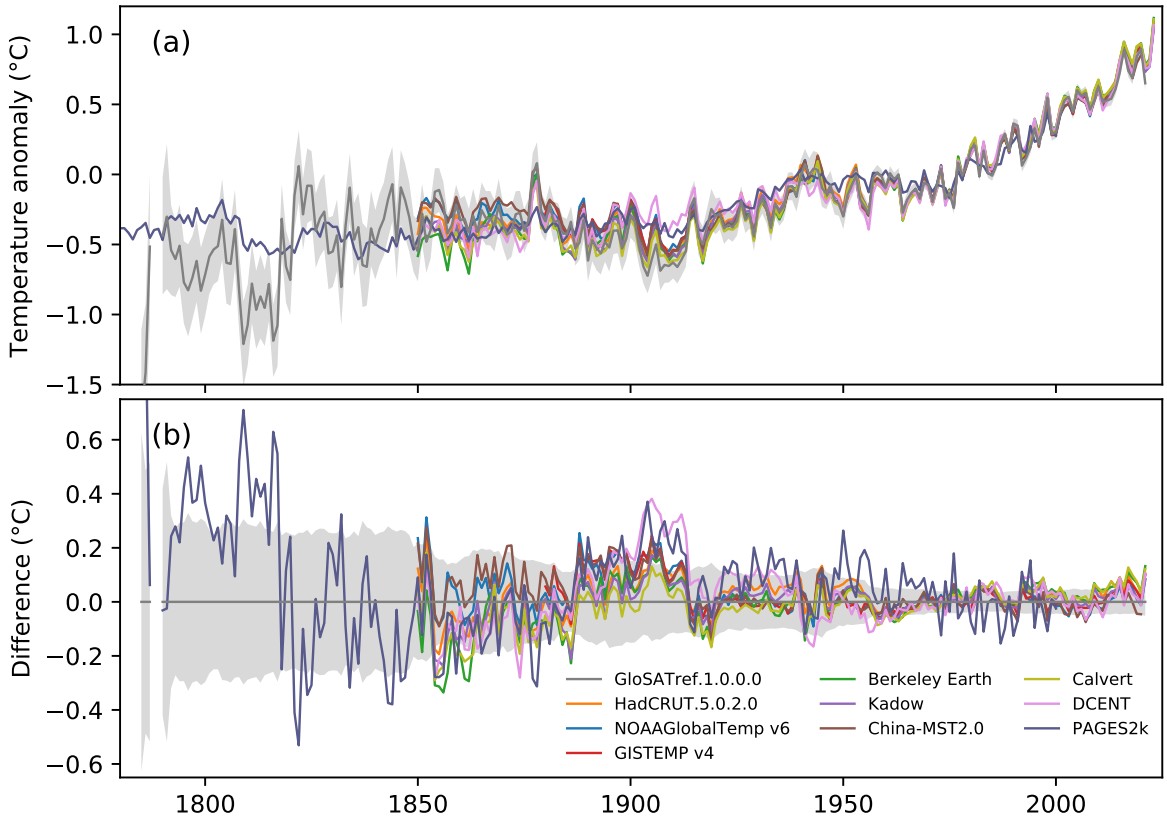

**Figure 5.** (a) GSAT and GMST (°C relative to 1961-1990) derived from instrumental data sets and the ensemble median of the PAGES2k palaeoclimate reconstructions and (b) differences between GloSATref and instrumental GMST and the PAGES2k ensemble median (each series minus GloSATref). Solid lines show the ensemble means or medians where the dataset provides an ensemble; grey shading is the 2.5% to 97.5% confidence interval for the GloSATref series only. All instrumental series are January-December annual averages. PAGES2k series are representative of April-March annual averages.

variability in the GloSATref global series is overestimated by reduced global measurement sampling (Fig. 2), which increases uncertainty and variance in the global-mean estimate, and by larger uncorrected errors in underlying measurements taken from less standardised instruments and measurement platforms. Coverage uncertainty estimates for the GloSAT series (based on sub-sampling late 20th century and early 21st century ERA5 reanalysis fields to the observed locations) may not fully represent the uncertainty. HadCRUT5 and especially GloSATref (which provides the coolest estimates at that time) are both cooler than the PAGES2k ensemble median during the early 20th century period, in contrast to Chan et al. (2024).





# 5   Conclusions

A new global surface air temperature (GSAT) anomaly dataset has been produced, GloSATref. This is the first such dataset to
use marine air temperature (MAT) instead of sea surface temperature (SST) in combination with land surface air temperature
(LSAT). Use of MAT has allowed extension of the global measurement-based record back to 1784, with a land-only analysis
back to 1781, adding nearly 70 years when compared to datasets using SST as their marine component which start in 1850 or
later. The construction of a MAT-based dataset required several challenges to be overcome, most notably the derivation and
application of adjustments to account for warm biases due to daytime heating of the ship and sensor environment (Cropper
et al., 2023). Similarly, for air temperature observations made over land, new estimates of bias were required to be able to use
observations made before the wide-spread adoption of Stevenson-style screens to shelter the thermometers from the direct or
indirect effect of solar radiation (Wallis et al., 2024). These adjustments will also improve the existing LSAT record (Osborn
et al., 2021) before about 1930.

Extending the instrumental near surface temperature record earlier in time by nearly 70 years gives an estimate of temperature
changes associated with a period of strong volcanic activity between the 1780s and the 1820s, albeit without full global
coverage. These data are a new resource for study of the climate of this early instrumental period (e.g. as in Ballinger et al.
in prep.), and provide an additional line of evidence for understanding global near surface temperature change and variability
alongside existing GMST data sets.

Instrumental monitoring of GSAT is contingent on sustained observation of air temperature over ocean and land. At present,
the marine air temperature observing network is less robust than that for SST, with numbers of MAT observations having
declined since the 1990s (Kent and Kennedy, 2021). Similarly, the LSAT record is dependent on maintenance of meteorological
station networks and international data exchange. Support for international data infrastructure to meet the requirements of the
construction of long term climate records remains essential (Folland et al., 2001; Kent et al., 2019; Li et al., 2021). Equally
important is support for data rescue through imaging and digitisation of observations to improve global sampling and of
metadata to improve understanding of observing methods throughout the record (Brohan et al., 2009; Brönnimann et al., 2018;
Luterbacher et al., 2024).

Global datasets based on SST have been produced for more than 30 years and have improved as understanding of the characteristics of the observations become better understood and methods of data set construction advance. Similar improvements
can be expected with records based on MAT if resources permit. This new global surface air temperature analysis provides an
additional line of evidence of changes in global temperature alongside existing GMST data sets and reanalyses.

*Data availability.*   On acceptance of this paper GloSATref and its constituent datasets will be made available via the CEDA archive (https://archive.ceda.ac.uk/). For data access during review, GloSATref and GloSATLAT are available from the Met Office Hadley Centre at https://www.metoffice.gov.uk/hadobs/glosatref/ and CLASSnmat v2 is available from https://figshare.com/articles/dataset/CLASSnmat_v2/27015661?file=49299550.



**Data used in comparison plots**

- CRUTEM.5.0.2.0 (Osborn et al., 2021) was downloaded from https://www.metoffice.gov.uk/hadobs/crutem5/. Data set accessed 2024-01-10.

- CLASSnmat v1: Cornes, R.C.; Kent, E.C.; Berry, D.I.; Kennedy, J.J. (2020): CLASSnmat: monthly, global, gridded night marine air temperature data. Centre for Environmental Data Analysis, 2023-06-13. https://catalogue.ceda.ac.uk/uuid/5bbf48b128bd488dbb10a56111feb36a/

- HadCRUT.5.0.2.0 (Morice et al., 2021) was downloaded from https://www.metoffice.gov.uk/hadobs/hadcrut5/. Data set accessed 2024-01-10.

- HadSST.4.0.1.0 (Kennedy et al., 2019) was downloaded from https://www.metoffice.gov.uk/hadobs/hadsst4/. Data set accessed 2024-05-21.

- NOAAGlobalTemp v6: Huang, B., X. Yin, M. J. Menne, R. Vose, and H. Zhang, NOAA Global Surface Temperature Dataset (NOAAGlobalTemp), Version 6.0.0 subset used: aravg.ann.land_ocean.90S.90N.v6.0.0.202407.asc. NOAA National Centers for Environmental Information. https://doi.org/10.25921/rzxg-p717 https://www.ncei.noaa.gov/data/noaa-global-surface-temperature/v6/access/timeseries. Data set accessed 2024-09-10.

- GISTEMP v4: GISTEMP Team, 2024: GISS Surface Temperature Analysis (GISTEMP), version 4. NASA Goddard Institute for Space Studies. Data set accessed 2024-09-10 at https://data.giss.nasa.gov/gistemp/.

- Berkley Earth (Rohde and Hausfather, 2020) was downloaded from www.berkeleyearth.org. Data set accessed 2024-09-10.

- Kadow: Johannes Meuer, Étienne Plésiat, Naoto Inoue, Maximilian Witte, Stephan Seitz, & Christopher Kadow. (2024). FREVA-CLINT/climatereconstructionAI: v1.0.3b. Zenodo. https://doi.org/10.5281/zenodo.11262704. Version 6. HadCRUT5.anomalies.Kadow_et_al_2020_20crAI-infilled.ensemble_mean_185001-202312.nc. Data set accessed 2024-03-01.

- China MST2.0: China MST2.0-Imax was download from https://figshare.com/articles/dataset/China-MST2_0_datasets/16929427/4. Data set accessed 2024-09-10.

- Calvert: Calvert, B.T.T. (2024) Maximum likelihood estimates of temperatures using data from the Hadley Centre and the climate research unit (version 1.2). Hamburg, Germany: World Data Center for Climate (WDCC) at DKRZ. Available from: https://doi.org/10.26050/WDCC/HadCRU_MLE_v1.2. Data set accessed 2024-09-10.

- DCENT v1:Chan, Duo; Gebbie, Geoffrey; Huybers, Peter; Kent, Elizabeth, 2024, "DCENT: Dynamically Consistent ENsemble of Temperature at the earth surface", https://doi.org/10.7910/DVN/NU4UGW, Harvard Dataverse, V1. Data set accessed 2024-09-10.



– PAGES2k: Neukom, R.; Barboza, L.A.; Erb, M.P.; Shi, F.; Emile-Geay, J.; Evans, M.N.; Franke, J.; Kaufman, D.S.; Lücke, L.; Rehfeld, K.; Schurer, A.P.; Zhu, F.; Brönnimann, S.; Hakim, G.J.; Henley, B.J.; Ljungqvist, F.C.; McKay, N.P.; Valler, V.; von Gunten, L. (2019-07-24): NOAA/WDS Paleoclimatology - PAGES2k Common Era Surface Temperature Reconstructions. NOAA National Centers for Environmental Information. https://doi.org/10.25921/tkxp-vn12. Accessed 2023-04-14.

*Author contributions.* CPM wrote the first draft of the paper with contributions from ECK, RCC, NR, JJK, TJO, EH and APS. The processing and analysis of the land observations was performed by MT, EJW, KC and TJO, that for the marine observations by TC, RCC, DIB, BRR, PRT and ECK. CPM, JW, JJK and NR constructed the combined gridded analysis. All authors reviewed the final text.

*Competing interests.* None

*Disclaimer.* None

*Acknowledgements.* We thank David Lister and Phil Jones (Climatic Research Unit, UEA) for updates and acquisitions to the CRUTEM station database.

*Financial Support.* Authors were supported by the Global Surface Air Temperature (GloSAT) Natural Environmental Research Council (NERC) project under grants NE/S015647/2 (CPM DIB, RCC, TC, JJK, NR, JW, ECK), NE/S015582/1 (TJO, MT, EJW), NE/S015566/1 (KC), NE/S015698/1b (APS) and NE/S015574/1 (EH, PT). In addition CPM, JJK, NR and JW

were supported by the Met Office Hadley Centre Climate Programme funded by DSIT, BRR was supported by NERC Grant NE/R015953/1 and EH was supported by the National Centre for Atmospheric Science (NCAS).



**Table 2.** Summary of new acquisitions and updates applied to CRUTEM.5.0.1.0 to create the GloSAT land surface air temperature station database (sdb). Place and country names are as given by the data source.

| Region | No. series | Source | Details |
|---|---|---|---|
| Routine updates | | | |
| Global | 7810 | CLIMAT | Updated series for 2020-2021 |
| Australia | 112 | BoM | Updated ACORN series for 2019-2021 |
| Canada | 434 | Environment Canada | Updated series and improved homogeneity |
| Chile | 314 | Chilean Centre for Climate and Resilience Research | Updated series for 2016-2021 |
| Denmark, Faroe Islands, Greenland | 17 | Danish Meteorological Institute | Updated series for 2017-2020 |
| Iceland | 127 | Icelandic Meteorological Office and Trausti | Updated series and improved homogeneity |
| New Zealand | 7 | NIWA | Updated series for 2018-2021 |
| Russia | 604 | GHCN-Daily | Updated series for 2018-2021 |
| Switzerland | 13 | MeteoSwiss | Updated existing series |
| USA | 1218 | USHCN | Updated series and improved homogeneity |
| New acquisitions | | | |
| Global | 439 | NCEI World Weather Records (WWR) | New series not previously in CRUTEM sdb |
| Global | 149 | NCEI Monthly Climatological Data for the World (MCDW) | New series not previously in CRUTEM sdb |
| Canada | 346 | Environment Canada | New, homogenised series not previously in CRUTEM sdb |
| Germany | 14 | DWD | New series not previously in CRUTEM sdb |
| Improved series (e.g. earlier extensions, gaps filled, improved homogeneity) | | | |
| Global | 2618 | NCEI World Weather Records (WWR) | Gaps filled, especially for 2011-2016 |
| China | 322 | CMA | Replacements with improved homogeneity |
| France | 49 | MeteoFrance | Mostly post-1950 additions to existing series |
| Germany | 68 | DWD | Replacements with improved completeness |
| Switzerland | 14 | MeteoSwiss | Replacements with improved homogeneity |
| Global | 15 | Multiple sources (papers, archives) | Jersey (1894-2019), Dublin (Ireland 1831-2021), Reading (UK 1908-2019), Perpignan (France 1836-2021), Bordeaux (France 1851-2021), Paris (France 1658-2019), Gorkij (Russia 1881-1989),Tenkodogo (Burkina Faso 1951-1991), Cucuta (Colombia 1961-2021), Nassau (Bahamas 1811-2021), St Helena (1892-2021), Ascension (1923-2021), Armagh (UK 1796-2021), Tianjin (China 1890-2021), Antananarivo (Madagascar 1889-2021) |



**Table 3.** Sources of station normal information in CRUTEM5 and GloSATref. Climatology uncertainty models for data derived, WMO and extrapolated normal are described in Brohan et al. (2006) and Morice et al. (2012).

| Normals source indicator | Normals description | CRUTEM station count | GloSATref station count | Climatology uncertainty model used |
|---|---|---|---|---|
| 1 | Normals missing | 2656 | 738 | Not applicable |
| 2 | Estimated using previous infilling methods | 100 | 11 | Extrapolated |
| 3 | WMO | 25 | 1 | WMO |
| 4 | Calculated directly from observations | 7855 | 5806 | Data |
| 5 | Taken from previous data set version | 3 | 7 | Data |
| 6 | Estimated solely from LEK | N/A | 1742 | Extrapolated |
| 7 | From combination of data and LEK | N/A | 3568 | Extrapolated |
| **Total** | **All usable types** | **7983** | **11135** | |

**Table 4.** Parameters used to construct the GloSATMAT height ensemble.

| Parameter | Sampling distribution |
|---|---|
| Year: start of increasing heights around 1870 | Normal: mean 1870, standard deviation 3 years |
| Year: start of World War 2 heights | Uniform: 1939 ± 2 |
| Year: end of World War 2 heights | Uniform: 1946 ± 5 |
| Height: constant until around 1870 | Normal: mean 6 m, standard deviation 2 m |
| Height: reached at start of World War 2 | Normal: mean 12 m, standard deviation 2 m |
| Height: change at start of World War 2 | Normal: mean -1 m, standard deviation 2 m |
| Height: reached by 1973 | Normal: mean 16 m, standard deviation 2 m |





**Table 5.** Representation of uncertainty model components in LSAT and MAT error models.

| Analysis domain | Uncertainty term | Analysis input | Error structure |
|---|---|---|---|
| LSAT | Urbanisation bias | Ensemble | One sided piecewise trend; correlated between stations to represent uncertainty in large scale averages (Morice et al., 2012) |
| | Exposure bias | Ensemble | Two-sided early record bias reducing to zero in mid-20th century; correlated between stations to represent uncertainty in large scale averages (Morice et al., 2012) |
| | Homogenization error | Ensemble | Random step change model, independent between stations (Morice et al., 2012) |
| | Measurement error | Error covariance | Random error uncorrelated between stations/grid cells (Morice et al., 2012) |
| | Grid cell sampling error | Error covariance | Random error uncorrelated between stations/grid cells (Morice et al., 2012) |
| MAT | Height adjustment / stability uncertainty | Ensemble | Per-ship height uncertainty and per 5° grid-cell/10-day period stability uncertainty encoded into ensemble grids |
| | Climatology uncertainty | Not used | Not used |
| | WW2 bias | Error covariance | Per-ship/year uncertainty during the period 1942-1945 for ships from Decks 195 and 245 (Cornes et al., 2020) |
| | Diurnal adjustment error | Not used | Included in measurement random error |
| | Measurement random error | Error covariance | Random error uncorrelated between ship/grid cells (Cornes et al., 2020) |
| | Measurement bias error | Error covariance | Correlated uncertainty between ships/grid cells (Cornes et al., 2020) |
| | Grid cell sampling error | Error covariance | Per-ship errors encoded into grid error covariance matrices (Cornes et al., 2020) |



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
