# Peer review of "An observational record of global gridded near surface air temperature change over land and ocean from 1781"

_Earth System Science Data, 2024_

## Referee Comment (RC1)

**Review of 'An observational record for global gridded near surface air temperature over land and ocean from 1781' by Colin P. Morice and colleagues**

**Summary**

The paper describes a major new dataset of surface temperatures. It describes the construction methods and undertakes a comparison (albeit superficial) to a range of existing datasets. The effort is important and publishable. The particular novelty of a GSAT rather than GMST product is noteworthy and further helps sample structural uncertainty in global surface temperature estimation in important ways.

In an ideal world I would like to have seen a more substantive comparison to the range of existing products but I don't think it reasonable to demand this in this paper. Rather, I would encourage a more substantive intercomparison in future work. None of my comments amount to show stoppers and I would encourage acceptance for publication.

**Major comments**

1. The introduction feels unduly short. This in part is because material I would have expected in the introduction instead is in Section 2. I would personally merge sections 1 and 2 as it feels decidedly odd to a reader to have such a short introduction followed immediately by what very much feels like introductory materials. Section 2 contains what, as a naïve reader I expected to be in the introduction and it throws me a bit as a reader to be given this structure.

2. I am very confused by what Figure 3 shows. I think what is throwing me is the label on the colour bar. Should this not be Temperature difference GloSATref – HadCRUT5? Otherwise the text and the figure are just in conflict with one another? Regardless, work is required here for clarity to avoid very confused readers between the figure and text to unconfound this.

3. While the data availability is well noted I don't see anything regarding code. Is the code used going to be made available?

**Minor comments**

1. Given that climate models warm SAT more than SST consistently across CMIP5 and CMIP6 ensembles it feels odd to make no reference to the fact that your results imply the opposite behaviour in the closing sentence of the abstract.

2. To Table 1 and associated discussion you may add the new JMA analysis?

3. Regarding the paragraph starting line 112 this is a bit misleading potentially to an unwary non-expert reader. While Berkeley Earth and GHCNM do not have explicit accounting for exposure bias both will be adjusting for the impacts of exposure bias where their breakpoint detection algorithms are identifying breaks. So while these are not specifically designed to identify and quantify exposure biases the de facto result is highly similar to an adjustment for exposure biases. The first sentence kind of hints at this but in a way that perhaps understates the impact.

4. In line 226 perhaps define pentad explicitly on first use (five day period) for a general reader who may not have come across this temporal aggregation before and especially since pentad can often be used for e.g. 5-year periods?
5. The sentence lines 356-357 I don't think make grammatical sense and regardless is very hard to parse. Please can you revise so as to be clearer what you mean here?
6. On line 421-422 it is unclear what is meant by the first period should it be to 1881-1900?
7. Line 435 would it be better to say less robustly quantified because uncertain uncertainties feels a little bit odd.
8. Line 438 Africa, and northern […] (missing 'and')

---

## Referee Comment (RC2)

This manuscript, titled "An observational record of global gridded near surface air temperature change over land and ocean from 1781", a new gridded data set of air temperature change across global land and ocean extending back to the 1780s. This data set, called the GloSAT reference analysis, has two novel features: it uses marine air temperature observations rather than the sea surface temperature measurements typically used by pre-existing data sets, and it extends further into the past than existing merged land and ocean instrumental temperature records which typically estimate temperature changes from the mid-to-late 19th century onwards. This study is important and interesting, and the results have practical implications for climate professionals. Therefore, I recommend accepting this article for publication on ESSD after some revisions.

1. It seems that the ESSD format requires the dataset's DOI to be included in the abstract section.

2. Section 3.1.2, "Handling of Early Exposure Bias," seems to be a good improvement, but from the text, we do not see the extent of this adjustment. It seems to still follow the approach of Morice et al. (2012)? In that case, there might be no need to elaborate so much on the method of Wallis et al. (2024); I suggest significantly condensing this section.

3. Section 3.1.3, "Treatment of Climate Normals," is also an advancement, but I am concerned about how different the new standard values' treatment is compared to the previous method, which used both standards climate normal and provisional normal. At the very least, a distribution of differences should be provided.

4. In Section 3.2, regarding the handling of ocean data, it is recommended to include a table that presents the differences brought about by the corrections and adjustments in each part.

5. In Section 3.3.4, The article mentions that sea ice regions are treated as land for weighting purposes. However, given that the extent of sea ice varies over time, could the authors provide further clarification on how the climatology is handled when calculating anomalies? Specifically, as the sea ice extent changes over time, regions may be influenced by both LSAT and MAT at different points, which could affect the calculation of climatology. Is there any method to adjust for or account for these dynamic changes to ensure the accuracy of the climatology calculations?

6. From Figure 4, the LSAT component of GloSATref has a slight cold bias compared to CRUTEM5 prior to 1950, while the bias in the MAT component is more pronounced. Between 1900 and 1960, the cold bias is more prominent (including GloSAT). Recently, a paper in Science pointed out that all existing SST datasets exhibit a significant cold bias before 1930, which is also reflected in GMST. However, in this study, GloSATref seems to show a certain degree of cold bias as well. Why? Given that SST exhibits a cold bias and GloSATref uses SAT, why is there still such a negative bias? This seems difficult to explain.

7. Similarly, Figure 5 also shows that the cold bias of GloSATref between 1900 and 1930 is more prominent.

---

## Author Comment (AC1)

**Response to reviewers for essd-2024-500 (An observational record of global gridded near surface air temperature change over land and ocean from 1781, by Morice et al.)**

**1. Referee No. 1**

**1.1 Summary**

The paper describes a major new dataset of surface temperatures. It describes the construction methods and undertakes a comparison (albeit superficial) to a range of existing datasets. The effort is important and publishable. The particular novelty of a GSAT rather than GMST product is noteworthy and further helps sample structural uncertainty in global surface temperature estimation in important ways.

In an ideal world I would like to have seen a more substantive comparison to the range of existing products but I don't think it reasonable to demand this in this paper. Rather, I would encourage a more substantive intercomparison in future work. None of my comments amount to show stoppers and I would encourage acceptance for publication.

**1.2 Referee No. 1 Major comments**

**1.2.1 R1 Major comment 1**

The introduction feels unduly short. This in part is because material I would have expected in the introduction instead is in Section 2. I would personally merge sections 1 and 2 as it feels decidedly odd to a reader to have such a short introduction followed immediately by what very much feels like introductory materials. Section 2 contains what, as a naïve reader I expected to be in the introduction and it throws me a bit as a reader to be given this structure.

Response: We have merged the previous section 2 into the introduction as suggested.

**1.2.2 R1 Major comment 2**

I am very confused by what Figure 3 shows. I think what is throwing me is the label on the colour bar. Should this not be Temperature difference GloSATref – HadCRUT5? Otherwise the text and the figure are just in conflict with one another? Regardless, work is required here for clarity to avoid very confused readers between the figure and text to unconfound this.

Response: We have revised the figure text and colour bar to correct this inconsistency. We have also reversed the difference shown in this figure. All difference plots in the paper are now consistent in showing differences as other data sets minus GloSAT.

**1.2.3 R1 Major comment 3**

While the data availability is well noted I don't see anything regarding code. Is the code used going to be made available?

This combines the results and processing workflows from previous studies without implementing new methods, and so we have not provided code. The methods used to construct the gridded analyses of LSAT, MAT and their combination, are described in previously published papers (notably the ensemble spatial analysis from Morice et al. (2021), MAT diurnal adjustments in Cropper et al. (2023), MAT gridding and uncertainty in Cornes et al. (2020), land station climatology calculation in Taylor et al. (2025), and early station exposure adjustments in Wallis et al. (2024)).

**1.3 Referee No. 1 Minor comments**

**1.3.1 R1 Minor comment 1**

Given that climate models warm SAT more than SST consistently across CMIP5 and CMIP6 ensembles it feels odd to make no reference to the fact that your results imply the opposite behaviour in the closing sentence of the abstract.

Response: As noted by Reviewer 3 (minor comment 5) we already included a short discussion of the behaviour of recent warming in GSAT and GMST in Section 4.3. We have not added this information to the abstract as it is hard to do so without providing the necessary context and physical understanding of SST/MAT warming differences.

**1.3.2 R1 Minor comment 2**

To Table 1 and associated discussion you may add the new JMA analysis?

Response: We have added COBE-STEMP3 analysis Ishii et al. (2025) to Table 1 and to the global time series comparison plot in Figure 5. Accompanying text has been added describing early record differences between DCENT and COBE-STEMP3 and other data sets reading:

"Differences over 1888 to 1913 (Fig. 5) are most prominent for the DCENT and COBE-STEMP3 data sets (Chan et al., 2024; Ishii et al., 2025), with each notably warmer than GloSATref over 1888 to 1913 and exhibiting excursions beyond the GloSATref 95% confidence intervals throughout the first half of the 20th century. These two recently published data sets have new treatments of inhomogeneities. Each uses land station LAT in their respective SST bias adjustment schemes. While the specifics of their adjustment methods differ, each method acts to reduce differences between detrended global average SST and LSAT anomalies. These two data sets also add SST data-source-based adjustments that impact early 20th century warming: COBE-STEMP3 includes an adjustment to address a truncation error in a prominent underpinning data source in this period, the KOBE collection (Chan et al., 2019), while DCENT uses an SST adjustment scheme based on nation, ICOADS deck and measurement method (Chan and Huybers, 2021)"

**1.3.3 R1 Minor comment 3**

Regarding the paragraph starting line 112 this is a bit misleading potentially to an unwary non-expert reader. While Berkeley Earth and GHCNM do not have explicit accounting for exposure bias both will be adjusting for the impacts of exposure bias where their breakpoint detection algorithms are identifying breaks. So while these are not specifically designed to identify and quantify exposure biases the de facto result is highly similar to an adjustment for exposure biases. The first sentence kind of hints at this but in a way that perhaps understates the impact.

Response: We have expanded these opening sentences to be clearer. The new text reads: "Current global data sets do not explicitly account for exposure bias across their networks of station data. Some (e.g. Berkeley Earth land record or those using GHCNm v4; see Table 1) rely on their general statistical homogenisation algorithms to detect and adjust for breakpoints arising from exposure changes. These are not specifically designed for identifying exposure biases, but where those biases are detected then they may yield a similar outcome. However, the power of breakpoint detection is reduced if changes in exposure were introduced within the same timeframe across many stations within a country or region (Menne and Williams, 2009), which was often the case for the introduction of Stevenson screens (Wallis et al., 2024). Other datasets leave exposure biases mostly uncorrected and instead represent their effects via a component in their uncertainty model, developed from previous exposure bias assessments (Brohan et al., 2006; Morice et al., 2012).

**1.3.4 R1 Minor comment 4**

In line 226 perhaps define pentad explicitly on first use (five day period) for a general reader who may not have come across this temporal aggregation before and especially since pentad can often

be used for e.g. 5-year periods?

Response: Pentad was already defined as 5 days on line 225 in the original manuscript, no changes made.

1.3.5 R1 Minor comment 5

The sentence lines 356-357 I don't think make grammatical sense and regardless is very hard to parse. Please can you revise so as to be clearer what you mean here?

Response: Edited for clarity as suggested

**1.3.6 R1 Minor comment 6**

On line 421-422 it is unclear what is meant by the first period should it be to 1881-1900?

Response: text edited for clarity to now state "in the 20-year averages from 1821-1840 to 1881-1900"

**1.3.7 R1 Minor comment 7**

Line 435 would it be better to say less robustly quantified because uncertain uncertainties feels a little bit odd.

Response: Suggested change made. The text now reads "the uncertainty estimates themselves are likely less robustly quantified".

1.3.8 R1 Minor comment 8

Line 438 Africa, and northern[...](missing 'and')

Response: Fixed

**2. Referee No. 2**

**2.1 Summary**

This manuscript, titled "An observational record of global gridded near surface air temperature change over land and ocean from 1781", a new gridded data set of air temperature change across global land and ocean extending back to the 1780s. This data set, called the GloSAT reference analysis, has two novel features: it uses marine air temperature observations rather than the sea surface temperature measurements typically used by pre-existing data sets, and it extends further into the past than existing merged land and ocean instrumental temperature records which typically estimate temperature changes from the mid-to-late 19th century onwards. This study is important and interesting, and the results have practical implications for climate professionals. Therefore, I recommend accepting this article for publication on ESSD after some revisions.

**2.2 Referee No. 2 Comments**

**2.2.1 R2 Comment 1**

It seems that the ESSD format requires the dataset's DOI to be included in the abstract section.

Response: Links to the data sets in the CEDA archive have been added to the data availability section. These web links are listed below, The below DOIs will be added to the manuscript once they become functional, linking to the corresponding CEDA webpages.

The data set files provided for the initial manuscript submission remain available from: https://www.metoffice.gov.uk/hadobs/glosatref/.

**The GloSATref analysis:**

*CEDA*: https://catalogue.ceda.ac.uk/uuid/a2519624a593402a83246bd359d098be/ *DOI*: https://dx.doi.org/10.5285/a2519624a593402a83246bd359d098be

**The GloSTLAT data set and station files:**

*CEDA*: https://catalogue.ceda.ac.uk/uuid/ef237f578329487eb02fb42f9db56bb2/ *DOI*: https://dx.doi.org/10.5285/ef237f578329487eb02fb42f9db56bb2

**The GloSATMAT data set:**

*CEDA*: https://catalogue.ceda.ac.uk/uuid/e6251bf935304cfbb9c9269dc7757a35/ *DOI*: https://dx.doi.org/10.5285/e6251bf935304cfbb9c9269dc7757a35

The "data used in comparison plots" section includes a new archive of the CLASSnmat v2 data set: CEDA: https://catalogue.ceda.ac.uk/uuid/306246329ae04eb3b2299446d911530a/ DOI: https://dx.doi.org/10.5285/306246329ae04eb3b2299446d911530a Figshare (provided at initial submission): https://doi.org/10.6084/m9.figshare.27015661

**2.2.2 R2 Comment 2**

Section 3.1.2, "Handling of Early Exposure Bias," seems to be a good improvement, but from the text, we do not see the extent of this adjustment. It seems to still follow the approach of Morice et al. (2012)? In that case, there might be no need to elaborate so much on the method of Wallis et al. (2024); I suggest significantly condensing this section.

Response: We have not condensed this section because we are adopting the exposure bias adjustments estimated by Wallis et al. (2024) which is different to the approach of Morice et al. (2012), and we give these details of what we did differently in the penultimate paragraph of this subsection. However, the final paragraph then notes that, to be conservative, we still retain the uncertainty term for exposure bias used by Morice et al. (2012). So the data are different, due to the applied Wallis et al. (2024) adjustments, but uncertainty ranges remain based on the Morice et al. (2012) approach. We have modified the text to make this point more clearly.

The extent of the adjustments can be see in differences between CRUTEM5 and GloSATLAT series in Figure 4(d) and are the primary cause of differences between HadCRUT5 and GloSATref analysis anomalies prior to the 1930s in Figure 3. The regions affected by exposure bias adjustments are indicated in Wallis et al. (2024) Figure 11, with seasonal affects shown in Figures 12 and 13 of Wallis et al. (2024).

**2.2.3 R2 Comment 3**

Section 3.1.3, "Treatment of Climate Normals," is also an advancement, but I am concerned about how different the new standard values' treatment is compared to the previous method, which used both standards climate normal and provisional normal. At the very least, a distribution of differences should be provided.

Response: Comparisons like those requested are included in Taylor et al. (2025), which is now under review with Geoscience Data Journal. We are happy to provide a preprint if requested to aid the review process of the current paper. To avoid duplication, we have not repeated these figures in the current paper, but we do provide two figures in the response to illustrate the quite close agreement between the old and new normals. First, where we have between 15 and 30 years of 1961-1990 data for a station, Fig. 1 shows that those estimated using local expectation kriging are in overall agreement with those calculated directly from the data, with only a few larger differences and very little bias in the median estimates. Second, we show that the new normals avoid the small bias that occurred previously when normals were computed from partially complete data that lie either early or late within the 1961-1990 baseline (Fig. 2). So this demonstrates a small improvement in the normals. The main improvement of course is that we now estimate normals for many more stations that previously had no normal estimated; we cannot show this improvement by a distribution of differences, since there are no old normals to compute the difference from. Again, there are figures in Taylor et al. (2025) showing the change in coverage arising from these newly estimated normals. No changes have been made to the paper in response to this comment.

Figure 1: Extract from Fig. 4 of the paper (Taylor et al., 2025) describing the new estimation of normals via local expectation kriging (LEK), that is under review with *Geoscience Data Journal*. Comparison of climatological normals calculated from LEK values with those calculated from co-located station data. Left: scatter plots of all values. Right: medians over all stations by month, with differences in medians below.

**2.2.4 R2 Comment 4**

In Section 3.2, regarding the handling of ocean data, it is recommended to include a table that presents the differences brought about by the corrections and adjustments in each part.

Figure 2: Extract from Fig. 8 of the paper (Taylor et al., 2025) describing the new estimation of normals via local expectation kriging (LEK), that is under review with *Geoscience Data Journal*. Comparison of climatological normals calculated from LEK values with those calculated from co-located station data for three example months (top: Jan; middle: May; bottom: Sep). The differences are shown as a function of mean year of available station data, with positive differences occurring when available data lie mostly within the early part of the 1961-1990 period and negative differences when available data are mostly later in the 1961-1990 period.

Response: It isn't possible to extract the impact of the adjustments applied from the interpolated dataset presented here. However timeseries and maps of the impact of the height adjustment are given in Cornes et al. (2020) and the approach taken means that these impacts will be the same in GloSATref. The marine air temperature heating bias adjustment of Cropper et al. (2023) brings the adjusted all hours MAT into close agreement with NMAT as shown in Figure 4e. We will investigate presenting the effect of the adjustments in the way suggested in relevant future papers but simple comparisons of the heating bias adjustment will be heavily dependent on diurnal sampling making interpretation difficult.

**2.2.5 R2 Comment 5**

In Section 3.3.4, The article mentions that sea ice regions are treated as land for weighting purposes. However, given that the extent of sea ice varies over time, could the authors provide further clarification on how the climatology is handled when calculating anomalies? Specifically, as the sea ice extent changes over time, regions may be influenced by both LSAT and MAT at different points, which could affect the calculation of climatology. Is there any method to adjust for or account for these dynamic changes to ensure the accuracy of the climatology calculations? Response: As described for HadCRUT5 in Morice et al. (2021) the land and marine components are analysed as anomalies separately and combined using weights for land/ocean fraction including time-varying sea-ice fraction. Other approaches are possible (e.g. Cowtan et al., 2015; Calvert, 2024) but our approach here was to follow the HadCRUT5 method for ease of comparison focusing on the impact of using MAT instead of SST as the marine component. No changes have been made to the paper in response to this comment.

**2.2.6 R2 Comments 6 & 7**

From Figure 4, the LSAT component of GloSATref has a slight cold bias compared to CRUTEM5 prior to 1950, while the bias in the MAT component is more pronounced. Between 1900 and 1960, the cold bias is more prominent (including GloSAT). Recently, a paper in Science pointed out that all existing SST datasets exhibit a significant cold bias before 1930, which is also reflected in GMST. However, in this study, GloSATref seems to show a certain degree of cold bias as well. Why? Given that SST exhibits a cold bias and GloSATref uses SAT, why is there still such a negative bias? This seems difficult to explain.

Similarly, Figure 5 also shows that the cold bias of GloSATref between 1900 and 1930 is more prominent.

Response: As the reviewer notes, the study of Sippel et al. (2024) showed and provided a plausible explanation for a cold bias in SST observations in the early 20th century. In this paper NMAT also shows a cold bias in this period. Similarly the cold period in MAT observations is probably more obvious than that in SST as the cold anomalies are slightly larger but shorter duration (about 20 years). It is possible that future work will show that the MAT anomaly is also an observation artefact, but of different origin, but a mechanism for a cool bias in NMAT/MAT data in this period (with MAT in GloSAT adjusted to NMAT equivalent) has not yet been identified. This is described in the text starting on (old) line 516 "This difference is notable because there is evidence ... yet GloSATref is colder still ...". No changes have been made to the text in response to this comment.

**3. Referee No. 3**

**3.1 Summary**

This paper is a valuable piece of work which describes a new global temperature data set which adds considerably to the diversity of available data sets, both because of its length and because of its use of marine air temperature. The approach of the paper is to draw heavily on existing papers for its methodology (in particular the Cornes et al., 2020; Cropper et al., 2023; Wallis et al., 2024, papers) and it is not my intention here to re-review the methods used in those papers; the main significance of the work described here is in merging these methods into a consistent data set. My comments are relatively limited and I expect the paper will be publishable with only modest changes, but I would like to see the author responses to the major comments and have therefore given a 'major revisions' rating.

**3.2 Referee No. 3 Major comments**

**3.2.1 R3 Major comment 1**

Figure 5 indicates a marked breakpoint c. 1820 in the difference series between GloSAT and Pages 2K, which is not commented on at all in the current text. If this is indeed a response to high volcanic activity (which seems a plausible hypothesis), do you have any comment on how volcanic activity might impact GloSAT differently to the proxy records used in PAGES 2k? (If it turns out that this points to an unresolved inhomogeneity in PAGES 2k, that might have implications for the IPCC AR6 assessment (Cross-Chapter Box 1.2) that warming from 1750 to 1850-1900 was 0.1 (-0.1 to 0.3) °C, as PAGES 2k was a major data source for that).

Response: We consider the "breakpoint" to be part of the large variability seen in GloSATref during this period which as we discuss is likely due to a combination of the sparse sampling and the effects of volcanic forcing, combined with a likely underestimate of the variance of the PAGES2k ensemble median. Because of the uncertainty in this period we do not believe there is sufficient evidence to support a conclusion that there is an issue with conclusions based on PAGES 2k in the IPCC. There is clearly more work to be done assessing all types of temperature records in this early instrumental period. We have revised this paragraph to more clearly explore these issues and to comment on the changes around 1820 that the referee mentions in the added text in Section 3.4.

**3.2.2 R3 Major comment 2**

I think the implications of the land/sea mix of data and how that has changed over time should be considered – land is warming faster than ocean so a lack of land data may create biases relative to a more spatially complete series. I note here that there is essentially no Southern Hemisphere land data before 1850, and only a limited proportion of Southern Hemisphere oceans is sampled in the earlier years (from Figure 2, this looks like it's essentially the Europe-to-south Asia via the Cape of Good Hope route), which has the potential to create sampling biases. The incomplete spatial sampling also potentially creates other issues too – e.g. one would not expect the ENSO signal in global temperatures in a data set with the coverage of GloSAT pre-1850 to be similar to the ENSO signal in more globally complete sets.

Response: We have added additional discussion of LAT/MAT sampling in the new Supplement Section S3 and Fig. S5, showing global and hemispheric diagnostics including: LAT, MAT and merged SAT averages, relative weighting of LAT and MAT in those averages and a comparison to merged SAT series computed by applying a fixed weight LAT/MAT weighting for all years (labeled Merged SAT\* in Fig. S5). These results are indicative of the effects of changing LAT/MAT weighting on average timeseries throughout the record. We acknowledge the limitation of these comparison to only provide information on the effect of relative LAT/MAT changes for observed regions. Comparison to paleo series in Section 3.4 and Fig S6 is also relevant. In particular, see the new text describing limitations of observation sampling in comparison to PAGES2k series in the final paragraph of Section

**3.4.**

Spatial sampling limitations likely would impact the representation of ENSO signals in the early record, however, we note that early record ENSO is not studied in our manuscript. If a following study were to use the GloSATref analysis to study pre-1850 ENSO then its authors would need to consider data sampling limitations, with caveats on early sampling stated in the manuscript and relevant coverage information provided in the pre-existing Fig. S1 and Fig. S3.

We have revised the following statement in our conclusions to more clearly state early sampling limitations: "Extending the instrumental near surface temperature record earlier in time by nearly 70 years gives an estimate of temperature changes associated with a period of strong volcanic activity between the 1780s and the 1820s, albeit requiring consideration of limitations in early global data coverage".

**3.3 Referee No. 3 Minor comments**

3.3.1 R3 Minor comment 1

L42 – it would probably also be worth mentioning the almost complete lack of pre-1850 Southern Hemisphere land data at this point (especially as that then becomes relevant to later discussion).

Response: added "and over land in the southern hemisphere"

3.3.2 R3 Minor comment 2

L226 - typo - should read 'at least one value'

Response: Fixed

**3.3.3 R3 Minor comment 3**

L381-383 – I'm a little surprised that MAT has a longer length scale than SST, do you have any idea why this might be the case?

Response: The spatial scales for MAT at large scales (5-degree monthly) are driven by the large scale atmospheric circulation and associated wind patterns. Whilst these large scale features are also important for SST, changes are also driven by air-sea interaction and moderated by much slower advection resulting in on-average shorter spatial scales for SST.

Parameter estimates by the maximum likelihood method used can also be sensitive to the observational uncertainty values used. Variance in observations is interpreted as temperature field variability when the provided observation uncertainty magnitudes are small or as observation error when observation uncertainty magnitude is large. Overly large/small prescribed observation uncertainty estimates would result in overly smooth/rough temperature field models with longer/shorter de-correlation length scale parameters. Hence, we can not rule out that differences in SST/MAT length scale parameters could result from misspecified uncertainty magnitudes or error correlation information.

**3.3.4 R3 Minor comment 4**

L408-409 – 'slightly reduced southern hemisphere coverage' – this could perhaps be elaborated on – as I understand it, there is almost no MAT data south of 40  $^{\circ}$ S outside of summer (presumably because most of the observing ship traffic, such as it is, in that region is vessels servicing Antarctic and sub-Antarctic bases). Does the concentration of observations in one part of the year create a bias we need to consider? Also relevant to discussion at L447-450.

Response: The comment on reduced coverage in both cases is relative to either HadSST4 or HadCRUT5 coverage, which is also low, at this time. In the first case this has been clarified.

**3.3.5 R3 Minor comment 5**

L488-493 – it could be mentioned here (from AR6 findings) that in reanalyses GSAT also warms faster than GMST. This discrepancy between observed and model-based data sets is intriguing and definitely worth further study (as the authors state at L492-493).

Response: Added "and atmospheric reanalyses" as suggested

**3.3.6 R3 Minor comment 6**

Section 4.4 – if possible, I think it would be useful for comparison to report a metric for GloSAT temperature change for 1850-1900 to 2011-2020, to match the corresponding metrics reported for GMST data sets in AR6.

Response: We have not included this metric for two reasons (1) an upcoming study will more comprehensively compare long term warming than we can here and (2) we would also need to roll in additional uncertainty analysis for the decadal averages for uncertainty estimates to be comparable to those in AR6

**3.3.7 R3 Minor comment 7**

L525-537 – I would have expected that a lot of the additional variability in GloSAT comes from the incomplete (and spatially uneven) sampling pre-1850. I think that should be commented on before the reference to volcanic activity.

Response: This paragraph has been revised in response to another comment; in the revision, sparse sampling is now mentioned before volcanic activity.

**4. Comment by Zeke Hausfather**

Really excited about this new paper extending the temperature record further back in time. One small recommendation for the authors is to show how their pre-1850 land temperature reconstruction compares to that published by Berkeley Earth (which also goes back to the 1700s).

Response: We have added the comparison with the Berkeley Earth land data set to a new figure in the supporting nformation, Fig S7. The existing Fig. 4 in the main paper shows a set of closely related data sets to show the effects of incremental updates within GloSAT data sets compared to their predecessors. Adding Berkeley Earth land, with its different processing methods, to that figure would require a large increase in y-axis range that would obscure seeing the effects of incremental changes in GloSAT. However, we feel this is a valuable addition to the paper and so we have added comparisons to additional land and ocean data sets in Fig S7 and Fig S8 the supporting information. We have added the following text to direct readers to the added figure:

"Global average anomaly time series for GloSATref, GloSATLAT and GloSATMAT are shown in Fig.4 in comparison to methodologically related data sets, showing the effects of data and methodological changes contributing to GloSATref. Additional land and marine data set comparisons are shown in Supplement Fig. S7 and S8 including a broader range of data sets."

**5. Comments by Robert Rhode**

**5.1 RR comment 1**

I've attached a file with Berkeley Earth's Land time series overlaid on the manuscript's Figure 4a. There are obvious similarities, including the cold swings during the early periods with high volcanic activity. As an alternative instrumental reconstruction prior to 1850, I would agree [with the comment by Zeke Hausfather 4] that a comparison like this probably ought to be included in the paper.